https://doi.org/10.1038/s41467-019-08531-4　　**OPEN**

# Conformations and cryo-force spectroscopy of spray-deposited single-strand DNA on gold

Rémy Pawlak [1], J.G. Vilhena [1,2], Antoine Hinaut [1], Tobias Meier [1], Thilo Glatzel [1], Alexis Baratoff[1], Enrico Gnecco[3], Rubén Pérez [2,4] & Ernst Meyer[1]

Cryo-electron microscopy can determine the structure of biological matter in vitrified liquids. However, structure alone is insufficient to understand the function of native and engineered biomolecules. So far, their mechanical properties have mainly been probed at room temperature using tens of pico-newton forces with a resolution limited by thermal fluctuations. Here we combine force spectroscopy and computer simulations in cryogenic conditions to quantify adhesion and intra-molecular properties of spray-deposited single-strand DNA oligomers on Au(111). Sub-nanometer resolution images reveal folding conformations confirmed by simulations. Lifting shows a decay of the measured stiffness with sharp dips every 0.2–0.3 nm associated with the sequential peeling and detachment of single nucleotides. A stiffness of 30–35 N m$^{-1}$ per stretched repeat unit is deduced in the nano-newton range. This combined study suggests how to better control cryo-force spectroscopy of adsorbed heterogeneous (bio)polymer and to potentially enable single-base recognition in DNA strands only few nanometers long.

[1] Department of Physics, University of Basel, Klingelbergstrasse 82, 4056 Basel, Switzerland. [2] Departamento de Física Teórica de la Materia Condensada, Universidad Autónoma de Madrid, E-28049 Madrid, Spain. [3] Otto Schott Institute of Materials Research, Friedrich Schiller University Jena, D-07742 Jena, Germany. [4] Condensed Matter Physics Center (IFIMAC), Universidad Autónoma de Madrid, E-28049 Madrid, Spain. These authors contributed equally: Rémy Pawlak, J.G. Vilhena. Correspondence and requests for materials should be addressed to R.P. (email: remy.pawlak@unibas.ch) or to R.P. (email: ruben.perez@uam.es) or to E.M. (email: ernst.meyer@unibas.ch)

Nucleic acids (NA)[1] are among the most studied biomolecules nowadays due to their biological relevance, but also their nanodevice applications or computing[2–4]. Control over nucleotide sequences as well as knowledge of their folding properties has enabled the rational design of highly elaborate two- and three-dimensional DNA structures, the so-called "DNA origami"[5] programmed by Watson–Crick complementarity[6]. These remarkable advances have been made possible by the accurate determination of nucleotide characteristics beforehand, using noninvasive single-molecule manipulation techniques, such as optical tweezers[7–9] or magnetic tweezers[10,11]. Beside these approaches, force spectroscopy based on atomic force microscopy (AFM) also allows direct measurements of mechanical, adhesion[12], and tribological properties[13], as well as visualizing self-assembly processes. So far, such force spectroscopic experiments on biomolecules have been conducted under ambient conditions in solutions, mostly up to few tens of pico-Newton tensile loads. Mechanical properties are then dominated by thermal fluctuations and folding/unfolding of soft parts[14–17]. Only few AFM studies on long polymers strongly bound at both ends reached the nN force level where thermal fluctuations are largely suppressed[18]. To the best of our knowledge, no features attributable to sub-nanometer structural details have been observed in force versus extension curves recorded under ambient conditions.

Imaging of DNA has also been a long-term challenge. Although numerous groups successfully visualized DNA branches with impressive spatial resolutions in solution under ambient conditions[19–22], the highest accuracy has been reached using scanning tunneling microscopy (STM)/AFM imaging at cryogenic temperatures enabling reduced contamination[23–26]. These experiments further required efficient deposition techniques to successfully transfer the macromolecules from solution onto a substrate while maintaining UHV cleanliness standards[27–29]. However, the characterization of adsorbed biomolecules at the sub-nm level, specifically of DNA under such conditions[28], still remains rather unexplored. Notably, the recent advances in frequency modulation AFM[30] under cryogenic conditions have pushed spatial resolution of adsorbed molecules to the single-bond level[31], while force spectroscopy enables complex manipulations of single molecules at surfaces[32–36].

Here, we demonstrate that dynamic AFM-based force spectroscopy in cryogenic conditions (5 K) is a promising method for characterizing the mechanics of single-strand DNA (ssDNA) 20-cytosine oligomers down to the sub-nm level. Similar to the advent of cryo-electron microscopy for structure characterization of biosystems[37], further investigations along this line could open avenues toward the integration of DNA into solid nanodevices through biomechanical studies at this level of precision.

## Results

**Real-space imaging of spray-deposited ssDNA.** 20-Cytosine ssDNA oligomers were electrospray-deposited at room temperature on the precleaned Au(111) kept in ultrahigh vacuum (UHV). The surface was then annealed step by step up to a maximum temperature $T_{max}$ of 500 K. After each step, the resulting structures were subsequently imaged at 5 K using constant-current STM. As shown in Fig. 1a, the surface morphology evolves from large aggregates of several nanometers to 4-nm-long isolated oligomers. Clusters of various dimensions are observed by STM similar to previous results on ssDNA/Cu(111) deposited using a pulsed-injection technique[25,26]. These structures are too large (Fig. 1b) compared to the expected size of a single dehydrated ssDNA 20-mer (Fig. 1c). Before annealing, we therefore

suppose that a small amount of solvent molecules might also surround single oligomers.

To promote solvent desorption from the gold surface, we step-by-step annealed the surface that we later imaged by STM after each step (Fig. 1a). This results in a decreased apparent size of the "hydrated ssDNA clusters" ($T \approx 340$ K) as water molecules desorb from the surface. At $T \approx 440$ K, only "dehydrated ssDNA oligomers" are observed by STM with a length of about 4 nm (Fig. 1d). Their overall size also corresponds to the structure of a folded oligomer adsorbed on Au(111), as systematically predicted by our simulations performed under different adsorption conditions (see Supplementary Note 5) and superimposed top views in Fig. 1b, d, e. When $T_{max} \geq 500$ K, these dehydrated oligomers coalesce into several nanometer-long structures (Fig. 1e). Sub-nm contrast could be obtained along individual dehydrated oligomers and their assembly not only by STM, but also by using constant-height AFM with CO-terminated tips[31] (Fig. 1d, e). In spite of the potential resolution below the molecular level, the determination of the overall ssDNA conformation is rather difficult.

**Molecular dynamics simulations of ssDNA adsorption.** Instead of simulating the complex electrospray-deposition processes which start and end up in charge-neutral species (see Supplementary Note 1 and Supplementary Fig. 1), in all our simulations, we considered a single ssDNA oligomer together with charge-compensating counterions. For a comparison with Fig. 1b, a 20-cytosine ssDNA oligomer generated from the canonical B-form[1] (see Supplementary Note 2 and Supplementary Fig. 2) together with 19 Na$^+$ ions was inserted into a water droplet, equilibrated, and then allowed to adsorb onto an unreconstructed Au(111) surface at room temperature (see Supplementary Note 3). We first used the ssDNA conformation fully embedded in water (Supplementary Note 4 and Supplementary Fig. 4) that we relaxed only considering the first three surrounding hydration layers, i.e., within ~1 nm around the molecule corresponding to 1332 water molecules (see Supplementary Fig. 5). Within the first 10 ns of the simulation, the droplet size decreases due to the surface tension causing a considerable folding of the ssDNA chain into a compact structure of only 3-nm diameter (see Supplementary Fig. 5). The hydration layer and the ssDNA folded structure then remain stable during the rest of the 100-ns-long simulation. Note that much less folding was observed if the chain was completely immersed in water, having a total length of 6.4 nm (see Supplementary Fig. 4).

The droplet so prepared was placed 2 nm above the surface and let free to adsorb (Fig. 2a). At the first stage of the adsorption, a meniscus is formed between the hydration layer and the gold surface, whereas, after the 100-ns-long simulation, the ssDNA adsorbs folded directly on the gold surface, but is elsewhere surrounded by its hydration layer. Compared to the free droplet conformation, the ssDNA structure thus obtained is longer, which is in agreement with the STM images of hydrated oligomers on Au (111) (Fig. 1b). To test the effects of dehydration and thermal annealing on the final adsorption configuration, starting from the adsorbed hydrated ssDNA shown in Fig. 2a, we simulated (i) partial water evaporation by annealing at 450 K (stage 3a in Supplementary Fig. 6), (ii) the effect of removing all water molecules (stage 3b in Supplementary Fig. 7), and (iii) by simulating the adsorption of a single DNA strand at 400 K without any water molecules (stage 4 in Fig. 1c and Supplementary Fig. 7). All adsorption simulations, including stage 3 (hydrated ssDNA at 300 K) led to very similar folded adsorbed conformations ~4 nm in size, in good agreement with the prevalent experimentally observed structures after annealing at 440 K, as shown in Fig. 1.

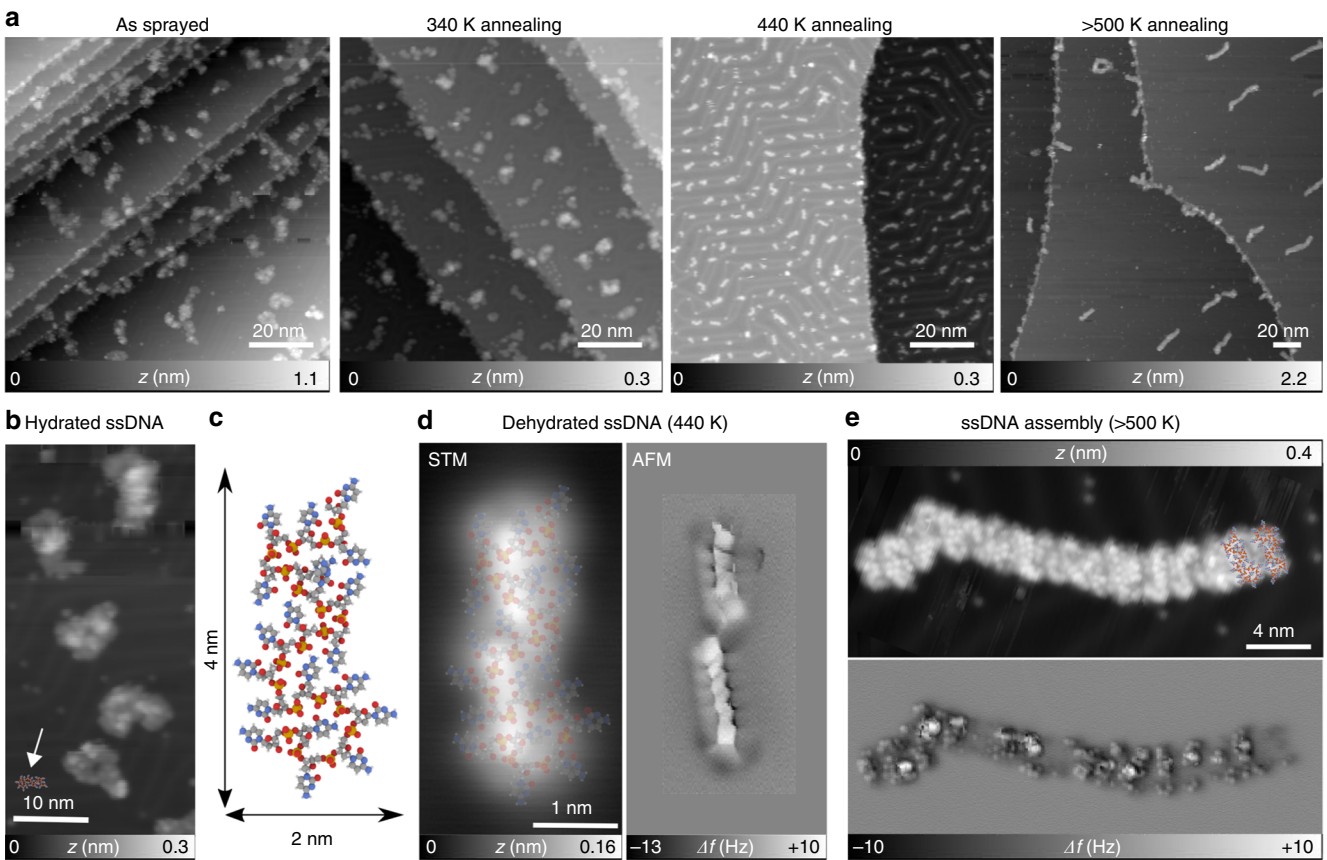

**Fig. 1** ssDNA morphologies as a function of Au(111) annealing temperature. **a** Overview of STM images of spray-deposited ssDNA on Au(111) at room temperature, after annealing at 340, 440, and above 500 K, respectively. **b** STM image of hydrated ssDNA after spray-deposition at room temperature. **c** Top view of a representative ssDNA structure on Au(111) obtained from MD simulations performed in vacuum at 400 K (see stage 4 in Supplementary Note 3). **d** STM image of a dehydrated single 20-cytosine ssDNA oligomer after 440-K annealing and the corresponding high- resolution constant-height AFM image, both acquired with a CO-terminated tip. The structural model of c is superimposed on the STM image. **e** STM image of self-assembled dehydrated ssDNA oligomers after 500-K annealing and the corresponding AFM image. All STM images were recorded at 5 K with $I = 2$ pA and $V = -1.3$ V. Properly scaled top views of the representative ssDNA structure are superimposed on **b**, **d**, **e** as a guide for the eye

The systematic nature of the findings can be traced back to the direct ssDNA/Au(111) contact already established at 300 K before dehydration. Concurrently with this direct contact, the interaction of cytosine bases with gold also induces a systematic flattening of the ssDNA structure. Indeed, most bases lie nearly parallel to the surface, similar to optimum adsorption structures for single-nucleotide bases on Au(111) computed in vacuum using a nonlocal van der Waals density functional[38], and very recently by MD simulations in vacuum and in water at 300 K[39].

The effect of thermally assisted diffusion of ssDNA on Au(111) has been investigated with 500-ns-long MD simulations of two adsorbed ssDNA at temperatures of 400 and 500 K (see stage 5 in Supplementary Note 3). Diffusion and coalescence of the ssDNA oligomers is observed only at $T = 500$ K (see Fig. 2b and Supplementary Note 6) which is in agreement with the experimental data. The oligomers preserve their folded adsorption characteristics during diffusion without showing significant flattening of the structure or unfolding. The side-by-side alignment of the molecules is again consistent with the STM/AFM images of Fig. 1c. The relatively high corrugation of the final MD-simulated structures confirms the difficulty to capture the ssDNA conformation from constant-height AFM images with CO-terminated tips (Fig. 1e).

**ssDNA cryo-force spectroscopy.** To investigate the mechanical properties of ssDNA adsorbed on Au(111), we have attempted to

lift single oligomers from the surface (Fig. 1c, d). We used the protocol introduced in refs. [33,40] to pull off single polyfluorene chains with the AFM tip. Experimentally, the tip was first gently indented into Au(111) to sharpen its apex, and then approached to one end of a selected oligomer until the tunneling current suddenly increased, thus indicating a jump to contact. A retraction curve was then recorded slightly beyond the position where the $I_t$ and $\Delta f$ dropped to their noise levels (see Supplementary Fig. 7). The yellow dots shown in the inset of Fig. 3b was the point at which the retraction process was initiated at a constant speed of $v = 22$ pm s$^{-1}$. The effective stiffness $k \approx 2k_0\Delta f/f_0$ ($k_0$ being the deflection sensor stiffness and $f_0$ its resonance frequency) decreases progressively from 23 to about 5 N m$^{-1}$ as the tip–sample separation Z increases. This variation is interrupted by narrow dips observed every 0.2–0.3 nm followed by an abrupt drop to zero when the tip is approximately at 1.4 nm far from the contact point, well below the length of one ssDNA oligomer in its adsorbed conformation (4 nm in Fig. 2). The premature detachment of the ssDNA from the tip (see Supplementary Notes 7 and 8) is confirmed by comparing STM images before and after the lifting experiment showing the entire oligomer still on the surface (inset Fig. 3b). Experimentally, only partial lifting of single ssDNA oligomers either self-assembled or individually adsorbed (see Supplementary Fig. 9) could be achieved. Thus, it appears that the folded ssDNA conformation and its strong adhesion on gold are the limiting factors of the experimental lifting. Indeed,

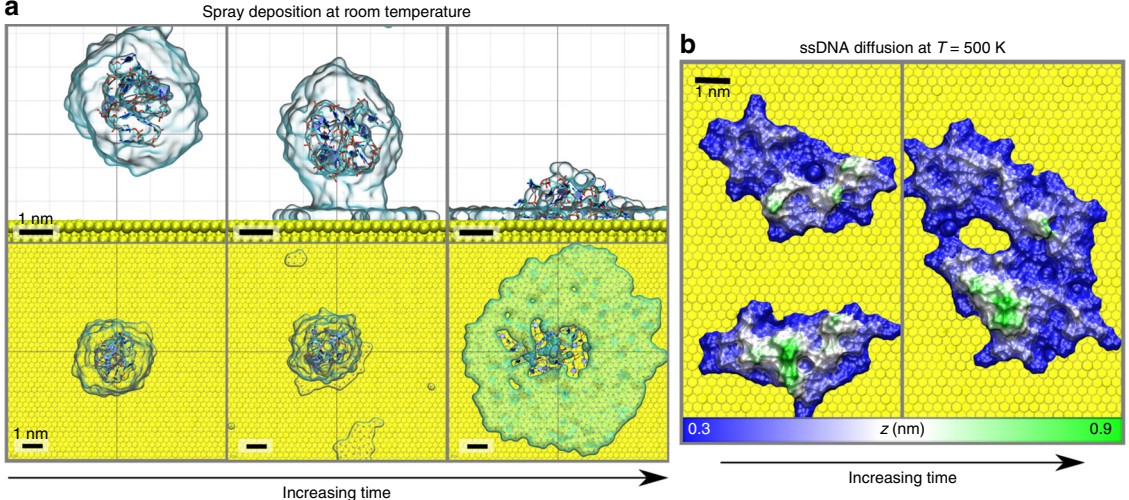

**Fig. 2** Molecular dynamics simulations of adsorption and diffusion on Au(111). **a** Side- and top views of a small water droplet containing one ssDNA oligomer getting adsorbed on the gold surface. Water is represented using a transparent surface. **b** Five-hundred nanosecond MD simulation of two oligomers assembled by diffusion at 500 K (stage 5 of the simulation protocol—see Supplementary Notes 3 and 6). At 500 K, both oligomers start diffusing, thus favoring self-assembly assisted by intermolecular interactions

desorbing a single-folded oligomer with the AFM tip requires not only peeling off the structure from the gold surface, but also unfolding part of the backbone prior to detachment. Each peculiarity of the folding configuration can thus cause an abrupt increase of the required force to lift the molecule and the rupture of the tip–molecule bond.

Steered MD simulations[41,42] have been run to shed light on the experimental results (see Methods and Supplementary Note 3). A representative theoretical retraction trace is shown in Fig. 3d. To do that, a virtual "tip atom" was connected to the P atom of the backbone between the first two nucleotides by a spring of stiffness $k_{tip}$ representing the tip–molecule bond and was pulled up at a constant speed of 0.1 m s$^{-1}$ at 5 K (Fig. 3a). From the recorded variations of the noise-averaged normal force $\langle F(Z) \rangle$ (Fig. 3c), the effective stiffness was extracted as $k = d \langle F(Z) \rangle / dZ$ (Fig. 3d), $Z$ being the distance between the tip atom and the pulled P atom at $t = 0$. In the simulations, the whole ssDNA oligomer detaches from the substrate when the tip has been retracted up to $Z_{off} = 11.8$ nm, which is slightly less than its fully stretched length (see Supplementary Fig. 8). In spite of the discrepancy with the experimental value of $Z_{off}$, the measured maximum $k$ values exhibit a similar trend in the common Z-range (blue area in Fig. 3d). Not only the stiffness $k$ decreases from comparable initial values of ~15–25 N m$^{-1}$, but pronounced dips (coinciding with abrupt force drops in the simulations) also appear at repeat distances of about 0.2–0.25 nm. Careful observation of the configurations adopted by the ssDNA atoms during simulated pulling reveals that the repeat distances reflect intermediate stages (peeling, lifting, and detachment) in the successive lifting of cytosine bases. These events are also accompanied by stick-slip-like sliding of the adjacent base over the Au(111) surface (0.28-nm lattice spacing) as well as irregular unfolding of the backbone. Details of such dynamics can be better visualized in Supplementary Movies 1 and 2. Note that the first step of the lifting process involves correlated base detachments and intricate unfolding, which require increased lifting forces. This increase might explain the premature detachment of ssDNA from the tip apex observed experimentally. Similar observations have been reported at room temperature in solution for grafted polymers[12] and ssDNA adsorbed on carbon nanotubes[17].

It is also remarkable that the computed $k$ variations as a function of $Z$ are very different from those previously reported for polyfluorene chains from Au(111) lifted with the same method[33]. There, the $k$ maxima (~0.4 N m$^{-1}$) were constant during retraction. The process ended at a distance corresponding to the number of monomers initially identified by STM on the surface. In that case (and also for graphene nanoribbons[34]), the much stiffer repeat units weakly adhere to the substrate, allowing nearly frictionless sliding and a complete chain detachment. The oligomer backbone is more flexible and most bases are strongly bonded to the gold surface. As shown in our MD simulation (snapshot of Fig. 3e and Supplementary Movies 1 and 2), the cytosine bases that remain adsorbed do not slide during lifting[33,34]. As a result, the lifted segment between the tip and the sample becomes essentially straight and inclined as the tip–sample separation increases. This sequential base detachment, similar to peeling off an adhesive tape, also reflects the strong adhesion of adsorbed cytosine bases. Interestingly, "infinite" as opposed to negligible friction was already detected for long ssDNA chains in solution at room temperature when adsorbed on gold[13] and graphite[15], respectively. Easy sliding of short ssDNA oligomers on graphene under cryogenic conditions was recently predicted by steered MD simulations[42].

The gradual reduction of the $k$ maxima arises because the stiffness of the lifted segment decreases as it becomes longer. Focusing on the most pronounced $k$ maxima achieved on the longest nearly linear parts of $F(Z)$, we assume local mechanical equilibrium in the springs-in-series model applied earlier to polyfluorene chains and to unzipped dsDNA hairpins. Including the stiffnesses $k_{tip}$ and $k_{pin}$ of the segment ends anchored to the tip and to the adsorbed part of the oligomer (Fig. 3e and Supplementary Fig. 9), the envelope of $k$ maxima is expected to satisfy

$$k = \left[ \frac{1}{k_{ends}} + \frac{n}{k_1} \right]^{-1}, \qquad (1)$$

with

$$\frac{1}{k_{ends}} = \frac{1}{k_{tip}} + \frac{1}{k_{pin}}. \qquad (2)$$

$k_1$ being the stiffness per repeat distance $b$ in fully stretched ssDNA and $n = \text{int} [Z/(b \cos \theta)]$ is the number of nucleotides detached from the substrate in the range where the lifted segments are straight and

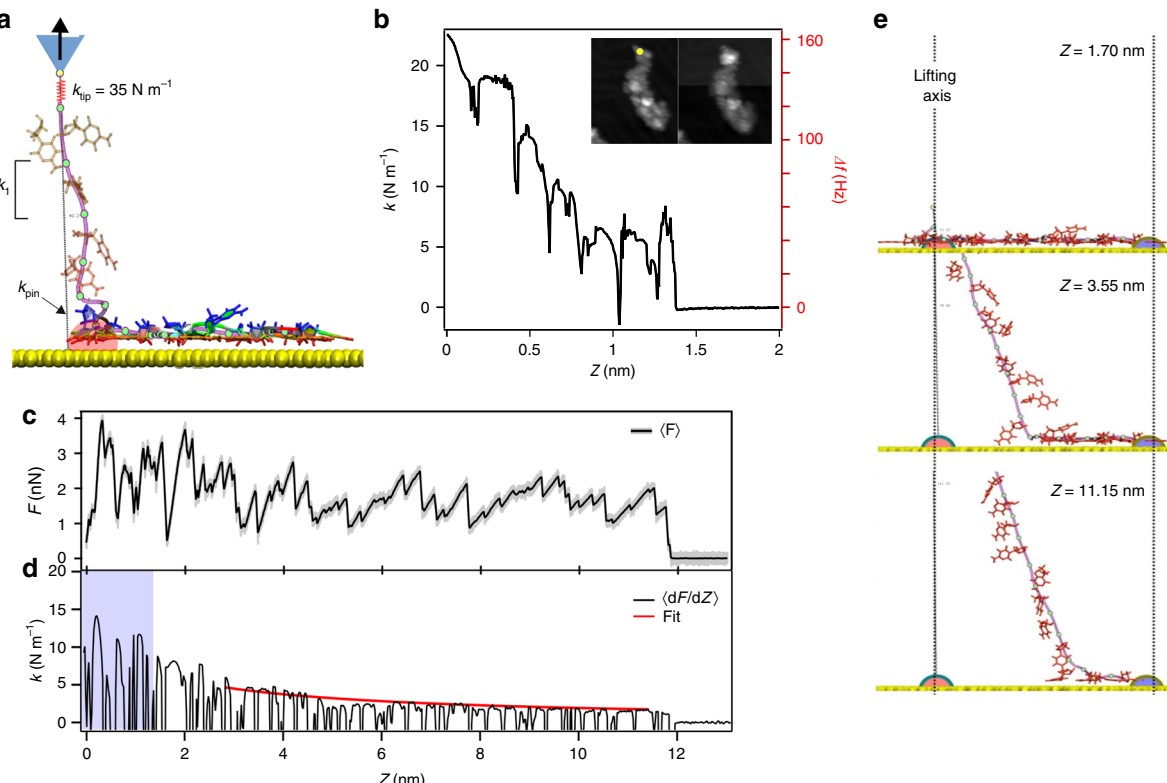

**Fig. 3** Mechanical response while lifting a ssDNA oligomer from Au(111). The oligomer is attached at one end to the AFM tip and pulled vertically at constant velocity. **a** Schematic of the lifting simulations. **b** Experimental retraction trace, $k(Z) \propto \Delta f(Z)$ recorded at 4.8 K. **c** Force-distance curve $F(Z)$ obtained from MD simulations assuming $k_{tip} = 35$ N m$^{-1}$ and **d** the corresponding computed stiffness $k(Z)$. The blue area shows the $Z$-range accessed by the experiment. The red curve shows the fit to $k$ maxima according to Eq. (1) with $k_1 = 32.6 \pm 3.9$ N m$^{-1}$, $k_{pin} = 18.3 \pm 3.3$ N m$^{-1}$, and $b \cos\theta = 0.64 \pm 0.05$. **e** Side views of the ssDNA oligomer after lifting 0, 10, and 17 nucleotides revealing the nearly straight configuration of the detached segment

inclined by $\theta$ (almost 18° between $n = 8$ and 18 according to Fig. 3e and the corresponding Supplementary Movies 1 and 2). The resulting fit (red curve) is superimposed on the computed $dF/dZ$ trace in Fig. 3d obtained for $k_{tip} = 35$ N m$^{-1}$. The resulting parameters are $k_1 = 32.6 \pm 3.9$ and $k_{pin} = 18.3 \pm 3.3$ N m$^{-1}$. This confirms that ssDNA is strongly adsorbed and that fully stretched ssDNA is much more compliant than polyfluorene[33], presumably because the stretched ssDNA backbone stiffness is dominated by bond angle bending. As a consequence, $k_{DNA} = k_1/19 = 1.7$ N m$^{-1}$ would be the stiffness of the fully stretched 20-cytosine oligomer with one base at each end subject to an average tension of ~2 nN. This load is one to two orders of magnitude larger than the maximum values attained in typical room-temperature investigations of ssDNA[12,14–17]. We obtained only slightly different results and fit parameters from independent simulations assuming higher and lower $k_{tip}$ values (see Supplementary Fig. 10). In particular, $Z_{off}$, $k_1$, $b$, and the average tension in the fitting range did not change appreciably.

The present work relies on closely matched scanning tunneling and force measurements, and computer simulations. The first part addresses the adsorption and self-assembly of single-strand DNA cytosine oligomers spray-deposited on Au(111) at the sub-nanometer level. Both theory and experiment showed folded ssDNA conformations arising from the first step of the adsorption. Furthermore, the unfolding of the ssDNA on the surface is not possible upon annealing. In the second part, the mechanical response of single ssDNA oligomers lifted from the gold surface is investigated at the same level using cryogenic force spectroscopy. Multistage detachment is inferred, similar to peeling off an adhesive tape, and reflects the strong adhesion of adsorbed ssDNA bases on gold. We extracted high values for the

initial stiffness measured during lifting (~15 N m$^{-1}$), as well as a comparable pinning stiffness obtained from numerical calculations and the intrinsic stiffness per repeat unit of fully stretched ssDNA (~33 N m$^{-1}$). This last value corresponds to the stiffness of the fully stretched 20-cytosine oligomer having a maximum length of 12 nm when subject to an average tension of ~2 nN. This load is one to two orders of magnitude larger than those applied at room temperature with single-molecule force spectroscopy on hundreds of nanometer-long ssDNA[12,14–17].

A drawback of the present system for experimental lifting is the complex folding of the ssDNA oligomers in comparison to polymeric systems in similar conditions[33,34]. This limits the lifting heights to only a fraction of the extended length. Compared to $k_{pin} \approx 0.7$ N m$^{-1}$ obtained for polyfluorene chains[33], a complete detachment of ssDNA might be achieved by using end linkers allowing to reinforce the bond between the oligomer and the AFM tip in UHV. Although more difficult than solution chemistry, this task appears feasible in view of the successful detachment of PTDCA from Au(111) following contact to a carboxylic oxygen atom[32]. In future experiments, we will focus on controlling the adsorption of the ssDNA using, for example, appropriately functionalized tips and patterned surfaces. We believe that such strategy might enable the complete lifting of linearly adsorbed oligomers and permit a meaningful statistical analysis of their detachment. Another alternative is to perform measurements on less adhesive surfaces[42]. Nevertheless, our results suggest that cryogenic force spectroscopy has the potential to study strongly adsorbed biomolecules or similar nano-sized synthetic systems with sub-nanometer resolution under tensile loads up to a few nanonewtons, i.e., 10–100 times higher than hitherto applied in most single-molecule force spectroscopy studies under ambient

conditions. Such studies might better characterize the diffusion properties or enable single-base distinction.

## Methods

**Sample preparation and electrospray deposition.** An Au(111) single crystal purchased from Mateck GmbH was cleaned by several sputtering and annealing cycles in a ultrahigh vacuum (UHV). The single-strand 20-mer DNA molecules were purchased from Microsynth (Switzerland). The solution provided was subjected to high-performance liquid chromatography (HPLC), thus ensuring that only the organic molecules present were 20-cytosine ssDNA oligomers. Subsequently, dialysis was performed in the presence of an excess of NaCl which guaranteed that any other salts generated during synthesis are exchanged by NaCl. The delivered solution contained 14.7 nM of ssDNA and a NaCl concentration sufficient to provide more than twice the required amount of $Na^+$ neutralizing counterions. This solution was further diluted to a concentration of around 1 nmol $ml^{-1}$ in water and then sprayed in UHV (see Supplementary Note 1). Depending on the applied voltage, the deposition time varies from 1 to 30 min at constant solution flux controlled by a syringe pump with a speed of $2-10 \times 10^{-6}$ l $min^{-1}$. Further details on the electrospray-deposition apparatus and characterization can be found in refs. [43–45].

**STM/AFM microscopy.** The STM/AFM experiments were carried out at ~5 K with an Omicron GmbH low-temperature STM/AFM controlled by a Nanonis RC5 electronics. We used commercial tuning fork sensors in the qPlus configuration[30], e.g., one prong-fixed, the other with an etched tungsten tip epoxied at the end (unperturbed frequency $f_0 = 26$ kHz, quality factor $Q = 10000–25000$ in UHV, and nominal spring constant $k = 1800$ N $m^{-1}$). These tips were sharpened by slight indentation into the gold surface; some were then terminated by a CO molecule at the apex picked up from the surface. All voltages refer to the sample bias with respect to the tip. The constant-height AFM images were acquired with CO-terminated tips using the noncontact mode by driving the free prong on resonance while maintaining a constant tip oscillation amplitude $A = 50$ pm. For such small oscillation amplitude, the frequency shift $\Delta f$ induced by a smoothly varying force acting on the tip is to a good approximation proportional to the gradient $k$ (effective stiffness) of the conservative force along the oscillation direction (perpendicular to the sample surface). The signal-to-noise ratio is then nearly optimal[30].

**Lifting experiments.** Pulling experiments were performed under the same conditions with gold-decorated tips while simultaneously recording the tunneling current at a typical bias of 40 µV. The ssDNA oligomers were picked up by gently pressing the AFM tip to the molecule at one of its extremities. Attachment of the molecule to the apex is revealed by an abrupt jump in the force and current signals, as shown in Supplementary Fig. 7. Force spectroscopic measurements upon retraction were performed at a velocity of 22 pm $s^{-1}$. In contrast to such measurements on biomolecules in ambient conditions, the gradient of the force along the oscillation direction rather than the pulling force itself is thus measured here.

**Atomic-level models and force fields.** In our simulations, we considered one Au(111) slab composed of three atomic layers-thick slab, where the positions of the atoms in the lowest layer are fixed during the MD runs using a harmonic restrain of 5 kcal $mol^{-1}$. Furthermore, we considered surfaces of two different sizes, i.e., $16 \times 16$ $nm^2$ (Fig. 2 and Supplementary Fig. 5) and $18 \times 22$ $nm^2$ (see Supplementary Fig. 6). The initial structure of the ssDNA molecule was generated using the software NAB[46], thus obtaining a double helix with the canonical B-form, as shown in Supplementary Fig. 2. Then, we removed the complementary sequence and used only one single-stranded 20-cytosine oligomer with the charged phosphate groups in the backbone of the ssDNA and 19 sodium counterions. The ssDNA atoms were described using both the parmbsc0[47] and the χOL3 refinements[48] of the Cornell ff99 force field[49]. The choice of this force field is motivated by its accuracy to describe the mechanical properties of DNA[50] as well as adsorption of biomolecules to surfaces[51]. The sodium counterions were described using the recently improved Joung/Cheatham parameters[52,53]. As for the gold atoms, we resorted to the CHARMM-METAL force-field[54,55] which simultaneously describes the intrinsic properties of gold, while retaining thermodynamical consistency with all the other force fields used here[54,55]. This force field has been extensively tested by studying the adsorption of different peptides (charged and uncharged) against both density-functional-theory simulations as well as available experimental results[54,55]. In the simulations performed in water (Fig. 2 and Suppl. Inf. Fig. S5), the water molecules are explicitly modeled using the TIP3P force field[56].

**Molecular dynamic (MD) simulation details.** MD simulations were carried out using AMBER14 software suite[46] with NVIDIA GPU acceleration[57–59]. Periodic boundary conditions and Particle Mesh Ewald (with standard defaults and a real-space cutoff of 2 nm) were used to account for long-range electrostatic interactions. Van der Waals interactions were truncated at the real-space cutoff, and Lorentz–Berthelot mixing rules were used to determine the interaction parameters between different atoms. In all vacuum simulations, the volume of the system was kept fixed and the temperature was adjusted by means of a Langevin thermostat with

a damping rate of 1 $ps^{-1}$ ensuring fast thermalization with a minimal effect on the fast slip dynamics. The SHAKE algorithm was used to constrain bonds containing hydrogen, thus allowing us to use an integration time step of 2 fs. Coordinates were saved every 1000 steps. In all our simulations, we observed that the final configuration was stable as it did not change during the last 40 ns of simulations (which was corroborated by the low, i.e., <0.2 nm, root-mean-square deviation). In the steered MD simulation results shown in Fig. 3 and Supplementary Fig. 10, the conservative force $F(Z)$ was computed as $k_{tip} (Z - Z_P)$ and its thermal average $<F(Z)>$ was approximated by a running average over (100 ps), an interval adjusted to obtain a smooth dependence without distortions except close to force drops. The resulting $d<F(Z)>/dZ$ can be safely[41] compared to the measured effective stiffness $k$ between negative slips in the intervals where $k$ is positive.

**Simulation protocols.** In total, we have performed eight different MD simulations, each labeled as a stage and described in Supplementary Note 3 and Supplementary Fig. 3. Here we briefly outline each stage. Stage 1: ssDNA oligomer fully embedded in water at 300 K. Stage 2: ssDNA inside a water droplet in vacuum at 300 K. Stage 3: Adsorption of a water droplet containing one ssDNA onto a Au(111) surface at 300 K. Stage 3a: Water evaporation at 450 K from the same adsorbed droplet. Stage 3b: Evolution of a fully dehydrated ssDNA adsorbed on Au(111) in vacuum at 400 K. Stage 4: Adsorption of a single DNA strand in vacuum onto Au(111) at 400 K. Stage 5: Diffusion and self-assembly of two ssDNA molecules adsorbed on Au(111) in vacuum at 400 and 500 K. Stage 6: Lifting a ssDNA molecule adsorbed on Au(111) in vacuum at 5 K. In all simulations, 19 $Na^+$ counterions per ssDNA molecule were included to ensure overall charge neutrality.

## Data availability

The data that support the findings of this study are available from the corresponding authors upon reasonable request.

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

## Acknowledgments

A.B., A.H., and E.M. and R.P., T.G., and T.M. thank the Swiss National Science Foundation (SNF) and the Swiss Nanoscience Institute (SNI). The COST Action MP1303 is gratefully acknowledged. Ru. P. and J.G.V. thank the financial support of the Spanish MINECO (projects MAT2014-54484-P, MDM-2014-0377, and MAT2017-83273-R) and also the computer resources, technical expertise, and assistance provided by the Red Española de Supercomputación at the Minotauro Supercomputer (BSC, Barcelona). J.G. V. acknowledges funding from a Marie Sklodowska-Curie Fellowship within the Horizons 2020 framework. Dr. Marcin Kisiel is acknowledged for fruitful discussion.

## Author contributions

E.M., Ru.P., R.P. and J.G.V. conceived the experiments. A.H. performed the electrospray deposition. R.P. performed the STM/AFM measurements and lifting experiments. J.G.V. conducted the numerical calculations. R.P. and J.G.V. analyzed the data and co-wrote the manuscript with the help of E.G. and A.B. E.M., Ru.P., R.P., J.G.V., A.H., T.M., T.G., A.B. and E.G. discussed on the results and revised the manuscript.

## Additional information

**Competing interests:** The authors declare no competing interests.

