## [Peer Review File · Nature Communications]

Reviewers' Comments:

Reviewer #1:

Remarks to the Author:

This is a lovely paper on cryo-force spectroscopy of ssDNA on Au surfaces. The authors have a unique experimental set up and have produced wonderfully detailed information about peeling and stretching of ssDNA from Au surfaces. There are some hard new numbers about ssDNA backbone elasticity that will be useful in a number of models and applications. The close connection with MD simulations is well-done and helps to interpret the measurements in terms of structural features. The model for stiffness reduction due to sequential lift-off of cytosine bases is simple but convincing. It was a pleasure to read and I recommend publication.

Reviewer #2:

Remarks to the Author:

The reviewed manuscript by Remy and his coworker describes elegant and front-end combination of scanning tunneling / force measurements, and computer simulations. I would like to admire the authors for their achievement.

Specially, supplementary movies are convincing.

However, my biggest concern is sample preparation. No offence. After careful comparison with "Ref26" (including related papers), my impression is that there may be a possibility of decomposition of 20-cytosine ssDNA oligomer.

Below I add comments and suggestions to the authors and editor that may improve the manuscript:

1. Figure 1a (a, STM overviews of spray-deposited ssDNA on Au(111) after annealing at 340 K, 440 K and above 500 K respectively ($I_t = 2$ pA, $V = -1.3$ V); scale bars = 20 nm.)

Fig.1a 340K, 440K annealing

My impression is that "20-cytosine ssDNA oligomers" appear to be too small for 20mer.

2. Fig.1a >500K annealing

It contains the same area as the image in Fig.6 Sd. It seems that the background subtraction has not been done for this STM image. If there is no particular reason, you should remove the background to make it easy to see.

3. Figure 1b (STM topographic image of a single 20-cytosine ssDNA oligomer after 440 K annealing ($I_t = 2$ pA, $V = -1.3$ V) and the corresponding higher resolution constant-height AFM image, both acquired with a CO-terminated tip; scale bar = 1 nm.)

My impression is that "20-cytosine ssDNA oligomer" appears to be too small for 20mer. It is difficult to imagine a corresponding molecular model. The z range of the STM image is 0.16nm. This suggest that the molecular image has an apparent height close to that of mono layer of base molecule. Is it possible to show a molecular model of 20-cytosine ssDNA oligomer lying flat on substrate? Or is it possible to reproduce the STM image out of 20-cytosine ssDNA oligomer using (semi empirical) theoretical calculation?

4. Figure 1c (c, STM topographic image of self-assembled sDNA oligomers after 500 K annealing ($I_t = 2$ pA, $V = -1.3$ V) and the corresponding AFM image; scale bar = 4 nm. The yellow line highlights the

length of a single ssDNA oligomer.)

The authors describe "the oligomers coalesce into several nano-meter long structures" It is very important to show a corresponding molecular structure model.

5. Fig.S1c-d (c-d, Profiles taken along the red dashed line in b of a of the topography $Z(X)$ and frequency shift $\Delta f(X)$.)

According to the STM profile, four peaks are seen; $X = 1.2, 2.1, 2.7, 3.3$ (nm).

$(3.3 - 1.2)/3 = 0.7$

The average interval between these peaks is 0.7 nm. This value is close to that described in ref26.

6. Fig.3b (Experimental retraction trace, $f(Z) / k(Z)$ recorded at 4.8 K.)

Does such a trace measure only once? I think that a histogram after multiple measurements will be convincing.

7. Fig.3b (Inset)

How do you know that the yellow spot is P atom or the end of ssDNA strand?

Reviewer #3:

Remarks to the Author:

Comments:

The paper by Pawlak et al. , "Conformations and cryo-force spectroscopy of spray-deposited single-strand DNA on gold" presents an experimental and computational investigation of the structure and properties of ssDNA on Au(111), exploiting cryogenic conditions to obtain a detailed description of the system. AFM experiments are used to obtain images of the DNA adsorption geometries on the gold surface at different temperatures, and to lift DNA from the surface to investigate its mechanical properties. Experimental results are supported by classical molecular dynamics calculations.

Although this work is potentially interesting as it is able to describe in details the interaction between DNA single strands and the gold surface, the presentation is confused and doesn't allow to really understand the importance of the obtained results.

Referring to the computational part in particular, technical details and results are split between the main text, the "Methods" Section and the "Supporting Information", and it is extremely difficult to have a clear picture of the different simulations performed, which are never properly listed. Some important details on the calculations performed are completely missing, as the dimension of the cell, the number of water molecules, the simulation temperature and the length of the simulations.

Due to this lacking of clarity and details, it is very difficult to evaluate the scientific contribution of the computational analysis and it would be impossible to reproduce it. In the main text, continuous references to the Supporting Informations interrupt the reading and suggest that the results are not well organized.

I strongly recommend the authors to reorganize the paper in a more clear and detailed way.

Major issues:

The Abstract focuses on the importance and innovation of AFM measurements of mechanical properties at low temperatures, that allow to overcome the thermal fluctuations noise which affects experiments performed at room temperature. However then the interest shifts from the technique applied to the system studied, without really explaining the importance of results obtained for DNA in the context of, for example, possible applications or further studies. The authors should emphasize which are the most important points of their work and discuss them.

In section "real-space imaging of spray-deposited ssDNA on gold" it is written that a water-DNA nano-droplet is virtually "sprayed" on the gold surface. In the Supporting Informations it is possible to understand that this term refers to a simulation in a cell with gold, DNA and some water molecules. This process and this terminology are not standard and a more extended description should be given.

Page 5, last lines: "Different initial ssDNA configurations with and without water molecules have also been considered" It is not clear if only the starting configurations were obtained with and without water, or if then also the adsorption process is simulated with and without water. Reading the Supporting Information doesn't fully clarify the point.

Page 6 first paragraph: "The effect of annealing on ssDNA/Au(111) has been also reproduced by MD simulations" and "Starting and final configurations are shown in Fig. 2B ... top views show that the oligomers essentially preserve the folded structure after self-assembly. Their alignment are consistent with the appearance of constant-height AFM images in Fig. 1c." The results of Figure 2b seems interesting but they are briefly discussed only in the supporting informations. The comparison with experimental structures is difficult as images are far in the text and the dimension a completely different. A Figure with a superposition of experimental and theoretical results could help the comparison.

Page 6: it is written that the DNA molecule detached from the AFM tip at 1.4 nm from the surface, and than it is stated that experimentally was not possible to achieve a complete lifting of the molecule. It is not clear if different measurements were performed with the molecule detaching from the tip at the same height (1.4 nm) or if a different height. There is something special about this length that forbids the lifting to proceed or this is the highest value of detachment reached?

On page 8, last paragraph, is presented a comparison with other works on molecules desorbed from surfaces with similar experimental procedures. The conclusions are reasonable, but the comparison is addressed in a disorganized and superficial way that should be improved. The same remark applies to the comparison discussed in the Conclusions.

In "ssDNA cryo-force spectroscopy" section, sentences are often very long without any punctuation and is very difficult to read and understand the text.

In the Methods section is written that CHARMM-METAL FF was used to describe the gold surface. This FF is not able to account for the dynamic polarization of the metal atoms in the Au surface. In the case of DNA, which carries a e charge for each nucleotide, this effect could be significant. Can the author comment about this point?

In Section S4 in the Supporting Information it is very difficult to understand which calculations are performed and many parameters are missing.

Figures S3 and S4 show the Na⁺ ions solvated in the large cell of water, but when water molecules are removed from the cell apart within 1nm from DNA, are Na⁺ ions all packed at this short distance

from the molecule? Does this choice affects the results of the simulation?

In paragraph "Simulated ssDNA diffusion and self-assembly on Au(111)" it is written "Note that a much longer simulation time was required". To do what?

Minor issues:

Figures of computational snapshot are small and printed with low resolution (e.g. Figure S3)

In the first panel of Figure S3 and S4 it is not clear what is the empty cube surrounding the DNA

Reviewers' comments:

Reviewer #1 (Remarks to the Author):

This is a lovely paper on cryo-force spectroscopy of ssDNA on Au surfaces. The authors have a unique experimental set up and have produced wonderfully detailed information about peeling and stretching of ssDNA from Au surfaces. There are some hard new numbers about ssDNA backbone elasticity that will be useful in a number of models and applications. The close connection with MD simulations is well-done and helps to interpret the measurements in terms of structural features. The model for stiffness reduction due to sequential lift-off of cytosine bases is simple but convincing. It was a pleasure to read and I recommend publication.

We sincerely thank referee#1 for reading our manuscript and acknowledge her/him for its positive comments.

Reviewer #2 (Remarks to the Author):

The reviewed manuscript by Remy and his coworker describes elegant and front-end combination of scanning tunneling / force measurements, and computer simulations. I would like to admire the authors for their achievement. Specially, supplementary movies are convincing. However, my biggest concern is sample preparation. No offense. After careful comparison with "Ref26" (including related papers), my impression is that there may be a possibility of decomposition of 20-cytosine ssDNA oligomer. Below I add comments and suggestions to the authors and editor that may improve the manuscript:

We thank referee#2 for carefully reading our manuscript and her/his comments. We answered to each points in the following and highlighted in red the changes in the attached pdf file of the manuscript.

The major concern of referee #2, comparing our results with those of ref. 26, is the possible degradation of ssDNA. Specifically, he refers to the differences in the size of ssDNA between our experiments and those in ref. 26. Below, we discuss these points in details.

Point#1: Deposition technique

Tanaka *et al.* work (ref26) have reported pioneering STM observations of short and long ssDNA strands adsorbed on a Cu(111) in ultra high vacuum (UHV). In that work (and publications from the same group), the deposition from solution was done with a *pulsed injection technique* where the opening of a small aperture is triggered by a high-speed valve. This way, droplets containing ssDNA oligomers are sucked into UHV preparation chamber and to the Cu(111) sample. The pulse

injection technique has the drawback to co-adsorb solvent molecules onto the surface since a partial pressures of molecules and solvent are generated during the valve opening, leading to an increased total pressure to about 10^{-7} mbars (ref 26).

Figure R1: **a**, STM images of the dye-terminated ssDNA 6-mers from ref. 26. **b**, STM image of 14 mers-ssDNA from Hamai *et al. J. Phys. Chem. B*, **2000**, *104*, 9894–9897. Particular features are attributed to (1) an isolated oligomer, (2) a shortened molecule, and (3) a cluster” of 15mers-ssDNA on Cu(111). **c**, STM image of 20 mers-ssDNA where a single oligomer (oval dashed line) has an approximate size of 4 nm.

In our work, we used an *electro-spray deposition* (ESD) apparatus (see Methods) with proven efficiency according to several experimental groups worldwide including ours. Over the past three years, we indeed published several papers on fragile molecular compounds adsorbed without decomposition at surfaces in UHV (see ref. 30). Other groups also showed the deposition of biological compounds with such apparatus without noticeable degradations (ref. 28-29). *In view of these results, we are confident that the degradation of the ssDNA oligomers during the spray-deposition is very unlikely.*

Thanks to the three differential pumping stages of the spray apparatus, a lower pressure in the UHV chamber can be reached compared to the pulse injection technique. A small amount of water molecules (well below a monolayer coverage) are nevertheless co-adsorbed. This is more the case with the pulse injection technique. This is explicitly mentioned in publications from Tanaka *et al* and their STM images shows solvent molecules (Fig. R1a). This is not a surprise since both deposition techniques, *e.g.* pulse valve and electro-spray deposition, use water as solvent for the ssDNA.

To further convince the referee that both techniques lead to similar deposition patterns at room temperature, in Figure R1, we show two STM images at 80K on Cu(111) from ref. 26 (Fig. R1a) and another two (b) from an earlier publication of the same group (new ref. 27) using different ssDNA oligomers of size closer than ours.

In Fig. R1a, short oligomers are observed among clusters. In Fig. R1b, longer ssDNA (14-mers) show features of various sizes attributed to single oligomers (intact or degraded) or to larger clusters. Finally, Fig. R1c shows STM images of 20mers-ssDNA. The size of the isolated 20-mers

encircled by the oval dashed line approximately 5 nm, is in excellent agreement with our STM images of "as-sprayed" 20-mers on Au(111).

Figure R2 shows a collection of such overview STM images acquired after ESD of 20-cytosines ssDNA molecules at room temperature and before surface annealing. Similar to Figure R1b, we observe features with sizes varying from 5 to 30 nm all over the surface. The apparent height in STM is about 0.2-0.3 nm which is in agreement with the observations of Tanaka *et al.*

Figure R2: Series of STM images of the 20-cytosine ssDNA spray-deposited on Au(111) at room temperature without annealing. **a-b**, Large scale STM image of the surface showing deposits of various sizes. **c**, Zoom of the hydrated ssDNA oligomers, the white arrows show co-adsorbed single H₂O water molecules. ($I = 1\text{pA}$, $V = -1.2\text{V}$).

In view of the referee comments, we decided to include those STM images in Figure 1 of the present version of the manuscript.

In the main text, we also discuss our results together with the instructive previous results of Tanaka *et al.* as follows:

“The oligomers were electro-spray-deposited at room temperature on the pre-cleaned Au(111) sample kept in ultra-high vacuum (UHV) at room temperature. The surface was then annealed step by step up to a maximum temperature T_{max} . After each step the resulting structures were subsequently imaged at 5 K using constant-current STM. As shown in Fig. 1a, the surface morphology evolves from large aggregates of several tens of nanometers to 4 nm long isolated oligomers. Clusters of various dimensions are observed by STM like in previous results on ssDNA/Cu(111) deposited using a pulsed-injection technique [26,27]. These structures are too large (Fig. 1b) compared to the expected size of a single dehydrated ssDNA 20-mer (Fig. 1c). Before annealing we therefore suppose that a small amount of solvent molecules might also surround single oligomers.”

Point#2: Effect of temperature on the ssDNA morphology: water desorption

The second step of the preparation consists in the annealing of the sample at different temperatures (Fig. 1). According to our numerical simulations (Fig. 2a), the ssDNA oligomers adsorb on Au(111) with their hydration layer. After spray-deposition without annealing, the observed clusters (Figure R2c) are single oligomers and/or self-assembled oligomers surrounded by water molecules. These “hydrated single-ssDNA” thus appear larger in the corresponding STM image than expected. Again, this is confirmed by the numerical simulations as shown in Fig. 2a. To better visualize the

dehydrated ssDNA structure, we included in Fig. 1b the ssDNA conformation obtained by MD simulations (shown by a white arrow).

By annealing the surface, we can promote the desorption of the water molecules from the gold surface. As a result, we first observed a decrease of the cluster size. At 440 K and above, we consider the single oligomers and their assemblies (>500 K) to be fully “*dehydrated*”. All crucial steps, i.e. nano-droplet deposition, water desorption, ssDNA diffusion and self-assembly were also confirmed by numerical simulations. Moreover, thermal degradation would not produce oligomers with such size distribution as shown in Fig. 1d and e. Here again, we are confident that degradation of the ssDNA oligomers during sample annealing is very unlikely.

Point #3: Length of the ssDNA in the STM/AFM images

According to our numerical simulations, a 20-mer folded in solution has a net length in the order of 6.4 nm (Supp Info Fig. S3). When confined into a “nano-droplet” as during electro-spray-deposition (Supp Info Fig S4), the “hydrated ssDNA” molecule in vacuum folds to an average diameter of ~3 nm. This is below the apparent length of adsorbed ssDNA oligomers observed in our STM experiments. Further molecular dynamic simulations predicts an average maximum length of 4-5 nm inside the droplet. In this case and also for dehydrated ssDNA discussed below, single oligomers are still folded but with most bases almost parallel to the surface. Nevertheless a few of the bases in the interior protrude, so that the final adsorption structures are nonplanar and disordered (Supp Info Fig S5).

To facilitate the comparison between experiment and theory, we superimposed over the STM images of new Figure 1 of the main text (see Figure R3 below) top-views of such an adsorption conformation on Au(111) obtained from our simulations. In the STM image b corresponding to spray-deposited ssDNA before annealing, the observed clusters are clearly too large to be naked single oligomers. As explained by our numerical simulations (Fig. 2 of the main text) these “hydrated ssDNA oligomers” on gold appear larger than expected.

In contrast, “*dehydrated single-oligomers*” imaged after annealing of the surface (Fig. 1d) perfectly match the simulated structure of Fig. 1c. In that case, the exact positioning of the bases and conformation of the ssDNA is difficult to interpret since the STM topography is influenced by the electronic density of the oligomer and not only its topography (as for the hydrated oligomers). Additionally, the corresponding AFM image shows only the topmost atoms of the molecular structure, what makes it difficult to interpret its exact conformation. In e, each units of the self-assembly also perfectly match the size of a single-dehydrated oligomer.

We commented on point #2 and #3 as follow:

“To promote solvent desorption from the gold surface, we step-by-step annealed the surface that we later imaged by STM after each step (Fig. 1a). This results in a decreased apparent size of the “hydrated ssDNA clusters” ($T \approx 340\text{K}$) as water molecules desorb from the surface. At $T \approx 440\text{K}$, only “dehydrated ssDNA oligomers” are observed by STM with length of about 4 nm (Fig. 1d). Their overall size also corresponds to the structure of a folded oligomer adsorbed on Au(111) predicted by our simulations and superimposed as top views in Fig. 1b, d and e. When $T_{\text{max}} \geq 500\text{K}$, these dehydrated oligomers coalesce into several nano-meter long structures (Fig. 1e). Sub-nm contrast could be obtained along individual dehydrated oligomers and their assembly not only by STM, but also using constant-height AFM with CO-terminated tips [32] (Figs. 1d and e). In spite of the potential resolution below the molecular level, the determination of the overall ssDNA conformation is rather difficult.”

Figure R3: Revised Figure 1 of the main manuscript. “Evolution of ssDNA morphologies as a function of Au(111) annealing temperature. **a**, Overviews STM images of spray-deposited ssDNA on Au(111) at room-temperature, after annealing at 340 K, 440 K and above 500 K, respectively. **b**, STM image of hydrated ssDNA after spray-deposition at room temperature. **c**, Top view of a representative ssDNA structure on Au(111) obtained from MD simulations. **d**, STM image of a dehydrated single 20-cytosine ssDNA oligomer after 440 K annealing and the corresponding higher resolution constant-height AFM image, both acquired with a CO-terminated tip. The structural model of **c** is superimposed on the STM image. **e**, STM image of self-assembled dehydrated ssDNA oligomers after 500 K annealing and the corresponding AFM image. All STM images were recorded at 5K with $I_t = 2 \text{ pA}$ and $V = -1.3 \text{ V}$. Properly scaled top views of the representative ssDNA structure is superimposed on **b**, **d**, and **e** as a guide for the eye.

1. Figure 1a (a, STM overviews of spray-deposited ssDNA on Au(111) after annealing at 340 K, 440 K and above 500 K respectively ($I_t = 2 \text{ pA}$, $V = -1.3 \text{ V}$); scale bars = 20 nm.) Fig.1a 340K, 440K annealing My impression is that "20-cytosine ssDNA oligomers" appear to be too small for 20mer.

We refer to our response to point #3 concerning the length of the ssDNA.

2. Fig.1a >500K annealing It contains the same area as the image in Fig.6 Sd. It seems that the background subtraction has not been done for this STM image. If there is no particular reason, you should remove the background to make it easy to see.

We thank the referee for noticing this mistake. We have now corrected the background (Fig. 1a) as in the supplementary figures.

3. Figure 1b (STM topographic image of a single 20-cytosine ssDNA oligomer after 440 K annealing ($I_t = 2$ pA, $V = -1.3$ V) and the corresponding higher resolution constant-height AFM image, both acquired with a CO-terminated tip; scale bar = 1 nm.)

My impression is that "20-cytosine ssDNA oligomer" appears to be too small for 20mer. It is difficult to imagine a corresponding molecular model. The z range of the STM image is 0.16nm. This suggest that the molecular image has an apparent height close to that of mono layer of base molecule. Is it possible to show a molecular model of 20-cytosine ssDNA oligomer lying flat on substrate? Or is it possible to reproduce the STM image out of 20-cytosine ssDNA oligomer using (semi empirical) theoretical calculation?

Here again, we refer to point#3 discussed above concerning the appearance and size of adsorbed ssDNA. The apparent STM height in our STM is 0.16 nm which is somewhat lower than in refs. 26 and 27. This relative height corresponds to a configuration with most bases almost parallel to the surface. To clarify this point as suggested by referee, top views of our structural model of the adsorbed ssDNA 20-mer is now included to the experimental STM image of Figure 1.

Regarding simulations of the STM image, the superimposed models are good enough to provide an estimate of the size of the image. A semi-empirical STM simulation is not going to provide reliable values of the apparent heights. This would require a sophisticated STM simulations that is beyond the scope of the present work.

4. Figure 1c (c, STM topographic image of self-assembled sDNA oligomers after 500 K annealing ($I_t = 2$ pA, $V = -1.3$ V) and the corresponding AFM image; scale bar = 4 nm. The yellow line highlights the length of a single ssDNA oligomer.) The authors describe "the oligomers coalesce into several nano-meter long structures" It is very important to show a corresponding molecular structure model.

As suggested by the referee, we added to the STM image of the assembly (Fig. 1e) top views of the representative structure above two ssDNA oligomers scaled with respect to the image size. Note that since the ssDNA conformation on Au(111) is complex, it is difficult to extract the exact positions and conformations of each cytosines as well as the ssDNA folded backbone from experimental images. The MD simulations illustrated in Fig. 3 and Fig. S6 address the self-assembly issue in details by investigating the diffusion of two ssDNA on gold upon annealing.

5. Fig.S1c-d (c-d, Profiles taken along the red dashed line in b of a of the topography $Z(X)$ and frequency shift $\Delta f(X)$.) According to the STM profile, four peaks are seen; $X = 1.2, 2.1, 2.7, 3.3$ (nm). $(3.3 - 1.2)/3 = 0.7$ The average interval between these peaks is 0.7 nm. This value is close to that described in ref26.

Our work, supported by molecular dynamic simulations, concludes that adsorbed ssDNA 20-mers lies nearly flat but folded on Au(111). This is in agreement with the experimental data as described in point#3 above. The length of the molecule at the surface is about 4 nm. A representative structural model is now shown in Fig.1c. As noticed by referee, the STM images shows protrusions spaced by 0.7 nm similar to some STM images in ref. 26. However, the corresponding AFM image shows much smaller spacing (about 0.2 nm). Therefore, we rather believe that the 0.7 nm spacing observed in our STM data results from a peculiar combination of electronic and structural effects rather than a clear evidence of the base position and of the inter-base spacing.

In addition, the inter-base distance of 0.65 nm reported in ref 26 corresponds to the conforamtion of a long ssDNA strand adsorbed on the Cu(111) in a straight fashion. This value is close to the

canonical B form contour length obtained in solution and to the average period of fully-stretched DNA obtained in our lifting simulations.

6. Fig.3b (Experimental retraction trace, $f(Z)$ / $k(Z)$ recorded at 4.8 K.) Does such a trace measure only once? I think that a histogram after multiple measurements will be convincing.

We measured only few approach-retract curves on different oligomers (hydrated and dehydrated) leading to similar conclusions, i.e. a characteristic decay of the $k(Z)$ curve as well as the observed 0.2-0.3 nm periods. The number of these curves is however not sufficient to make a meaningful statistical analysis at present. One important goal of the present manuscript is to provide a “*proof of concept*” for the characterization of the mechanical properties of DNA strands and of other heterogeneous (bio)polymers more resistant to pulling than those hitherto studied.

Although we believe that such studies would be interesting (similar to single-molecule force spectroscopy at room temperature), it will be challenging to collect enough data and converge to a clear conclusion. Indeed, numerous ssDNA conformations coexist on the annealed surface which will produce a large number of force spectroscopic data with differences depending on their folding. Prior to such statistical measurements, we think that it is more important to focus on a better control of the ssDNA attachment and folding using, for instance, patterned surfaces.

We therefore briefly discuss such future experiments in the Conclusion :

“In future experiments, we will focus on controlling the adsorption of the ssDNA using, for example, appropriately functionalized tips and patterned surfaces. We believe that such strategy might enable the complete lifting of linearly adsorbed oligomers and permit a meaningful statistical analysis of their detachment.”

Fig.3b (Inset) How do you know that the yellow spot is P atom or the end of ssDNA strand?

Experimentally, owing to drastically reduced drifts at low temperature we can control the spatial position of the tip with Ångström precision but neither know the exact conformation of the ssDNA molecules under the tip nor the attachment mechanism. Therefore, we cannot control the exact atom involved in pulling the ssDNA strand as in simulations. The yellow spot of the inset thus corresponds to the position of the tip where we acquired the approach-retract curve after a successful, yet not sufficiently strong attachment.

To clarify this point, we modified the text in the “ssDNA cryo-force spectroscopy” section as follow:

“Experimentally, the tip was first gently indented into Au(111) to sharpen its apex, then approached to one end of a selected oligomer until the tunneling current suddenly increased, thus indicating a jump to contact. A retraction curve was then recorded slightly beyond the position where the current and frequency shift dropped to their noise levels (Supp. Info Fig. S7). The yellow dot in the inset of Fig. 3b) shows the position of the tip where such a curve was acquired.”

Reviewer #3 (Remarks to the Author):

Comments:

The paper by Pawlak et al. , “Conformations and cryo-force spectroscopy of spray-deposited single-strand DNA on gold” presents an experimental and computational investigation of the structure and properties of ssDNA on Au(111), exploiting cryogenic conditions to obtain a

detailed description of the system. AFM experiments are used to obtain images of the DNA adsorption geometries on the gold surface at different temperatures, and to lift DNA from the surface to investigate its mechanical properties. Experimental results are supported by classical molecular dynamics calculations. Although this work is potentially interesting as it is able to describe in details the interaction between DNA single strands and the gold surface, the presentation is confused and doesn't allow to really understand the importance of the obtained results.

Referring to the computational part in particular, technical details and results are split between the main text, the "Methods" Section and the "Supporting Information", and it is extremely difficult to have a clear picture of the different simulations performed, which are never properly listed. Some important details on the calculations performed are completely missing, as the dimension of the cell, the number of water molecules, the simulation temperature and the length of the simulations.

Due to this lacking of clarity and details, it is very difficult to evaluate the scientific contribution of the computational analysis and it would be impossible to reproduce it. In the main text, continuous references to the Supporting Informations interrupt the reading and suggest that the results are not well organized.

I strongly recommend the authors to reorganize the paper in a more clear and detailed way.

We thank referee #3 for its comments helping us to improve the clarity of our manuscript. We carefully considered the comments and substantially revised the main text accordingly. Below, we address each point in detail and include the changes made in the main manuscript. These changes are highlighted in red in the attached pdf file.

Major issues:

The Abstract focuses on the importance and innovation of AFM measurements of mechanical properties at low temperatures, that allow to overcome the thermal fluctuations noise which affects experiments performed at room temperature. However then the interest shifts from the technique applied to the system studied, without really explaining the importance of results obtained for DNA in the context of, for example, possible applications or further studies. The authors should emphasize which are the most important points of their work and discuss them.

We are very grateful that the referee brought this to our attention. We changed the abstract accordingly, i.e. in a way that:

a) It naturally introduces the system of interest, i.e. DNA, as a prototypical example where structure and mechanical properties are both quintessential to understand its biological activity ; b) It better highlights the important points of this work, namely: the possibility to probe intramolecular forces in low thermal noise conditions and in a far unexplored force regime (nN), and to provide a first direct measurement of DNA mechanical stiffness at the base-pair level scale. Taking into consideration that most DNA-protein interaction occurs at the nanoscale, and also that this force regime allow us to explore the enthalpic deformations of ssDNA thus allowing us to better understanding the mapping between sequence-mechanical properties one may reasonably expect that this new approach may allow us to bridge the structural properties of DNA (obtained with cryo-EM, NMR or X-ray) with the entropic mechanical properties usually probed using optical/magnetic tweezers.

The Abstract has been modified as follow:

“Cryo-electron microscopy has become a valuable tool to determine the structure of biological matter in vitrified liquids. However, structure alone is insufficient to understand the dynamics and biological function of most biomolecules, often accompanied by mechanical actuations such as bending in chromosome packing or untwisting in DNA replication. So far, mechanical properties of biomolecules are essentially probed at room temperature using tens of pico-newton forces with a resolution usually limited by entropic thermal fluctuations. Here, we combine force spectroscopy and computer simulations in cryogenic conditions to quantify intra-molecular properties of spray-deposited single-strand DNA oligomers on Au(111). Images with sub-nanometer resolution reveal the folding conformations as confirmed by molecular dynamics simulations. Single-chain lifting shows a progressive decay of the measured stiffness with sharp dips every 0.2-0.3 nm associated with sequential peeling and detachment of single nucleotides. A stiffness of 30-35 N.m⁻¹ per stretched ssDNA repeat unit is measured in the nano-newton range. Cryo-force spectroscopy might thus provide new insights into adsorbed biological compounds under nano-Newton tensile stress and potentially enable single-base distinction in DNA strands—of only few nanometer long.”

In section “real-space imaging of spray-deposited ssDNA on gold” it is written that a water-DNA nano-droplet is virtually “sprayed” on the gold surface. In the Supporting Informations it is possible to understand that this term refers to a simulation in a cell with gold, DNA and some water molecules. This process and this terminology are not standard and a more extended description should be given.

As suggested by the referee, we have improved the description of our simulations both in the main text and the Supplementary Information..

The description of the spray-deposition simulation is now as follow:

“To elucidate the oligomer conformation, we reproduced using MD simulations the spray-deposition of one ssDNA on Au(111). For that, a 20-cytosine ssDNA oligomer generated from the canonical B-form [1] (Fig. S2) together with 19 charge-compensating Na⁺ ions was inserted into a water droplet and virtually “sprayed” onto an unreconstructed Au(111) surface at room temperature (Fig. 2a). To perform this simulation, we first used the ssDNA conformation fully embedded in water shown in Supp. Info Fig. S3 that we relaxed only considering the three first surrounding hydration layers, i.e. within ~ 1 nm around the molecule corresponding to 1332 H₂O molecules (Fig. S4). Within the first 10 ns of the simulation, the droplet size decreases due to the surface tension causing a considerable folding of the ssDNA chain into a compact structure of only 3 nm diameter (Fig. S4). The hydration layer and the ssDNA folded structure then remains stable during the rest of the 100 ns-long simulation. Note that much less folding was observed if the chain was completely immersed in water, having a total length of 6.4 nm (Fig. S3).

In Fig. 2a, the folded ssDNA conformation together with its hydration layer was virtually sprayed on gold by placing the droplet 2 nm above the surface and letting it free to adsorb. During adsorption, a meniscus is first formed between the hydration layer and the gold surface. At the final stage of the 500 ns-long simulation, the ssDNA adsorbs folded on the surface surrounded by its hydration layer. Compared to the droplet conformation, the structure became longer (~ 4 nm) in excellent agreement with the experimental images of Fig. 1b.”

Page 5, last lines: “Different initial ssDNA configurations with and without water molecules have also been considered” It is not clear if only the starting configurations were obtained with and without water, or if then also the adsorption process is simulated with and without water. Reading the Supporting Information doesn’t fully clarify the point.

The simulation of the ssDNA spray-deposition shown in Fig. 2a is done for the folded conformation of Fig. S4 with the hydration layer. To explore the effect of the initial conformations on the final adsorption, the canonical B form were also tested (Fig. S5). Given the impossibility of completely removing the water molecules from the simulation box during the dynamics, the adsorption of the ssDNA conformation previously obtained in the nano-droplet (Supp. Info Fig. S4) but without water molecules were also simulated by MD as shown in Supp. Info Fig. S5a.

To clarify the calculations, we modified the text as follow:

“Different initial ssDNA conformations have also been considered to test the effect of the hydration layer as well as the initial configuration on the final adsorption. Specifically, we simulated the adsorption of the folded ssDNA conformation without the hydration layer (Fig. S5a) and of the canonical B form without any solvent molecules (Fig. S5b). All cases always led to very similar folded conformations on gold.”

Page 6 first paragraph: “The effect of annealing on ssDNA/Au(111) has been also reproduced by MD simulations” and “Starting and final configurations are shown in Fig. 2B ... top views show that the oligomers essentially preserve the folded structure after self-assembly. Their alignment are consistent with the appearance of constant-height AFM images in Fig. 1c.” The results of Figure 2b seems interesting but they are briefly discussed only in the supporting informations. The comparison with experimental structures is difficult as images are far in the text and the dimension a completely different. A Figure with a superposition of experimental and theoretical results could help the comparison.

To clarify the comparison between experiment and theory suggested by the referee, we superimposed on the STM images shown in Figure 1 an exemplary ssDNA structure obtained from numerical calculations of Fig. 2b as a guide for the eye.

As suggested by the referee, we added details of the MD simulations of the ssDNA diffusion. We added to the main manuscript:

“The effect of surface annealing on ssDNA/Au(111) has been also simulated by running 500 ns–MD simulations of two adsorbed ssDNA at temperatures of 300 K, 400 K and 550 K, respectively. Starting and final configurations are shown in Supp. Info. Fig. S6.”

“The oligomers preserve their folded structure of their adsorption during diffusion without showing significant flattening of the structure or its unfolding. The side-by-side alignment of the molecules is consistent with the STM/AFM images of Fig. 1c. The relatively high corrugation in the MD simulations also confirm the difficulty to capture the ssDNA conformation by constant-height AFM images with CO-terminated tips.”

Page 6: it is written that the DNA molecule detached from the AFM tip at 1.4 nm from the surface, and than it is stated that experimentally was not possible to achieve a complete lifting of the molecule. It is not clear if different measurements were performed with the molecule detaching from the tip at the same height (1.4 nm) or if a different height. There is something special about this length that forbids the lifting to proceed or this is the highest value of detachment reached?

Lifting measurements have been performed few times on several ssDNA oligomers. Two typical curves are provided in our manuscript obtained for a single ssDNA just after spray-deposition (Fig. 3b) and after annealing of the surface (dehydrated conformation, Sup. Info Fig. S7). The highest

value reached experimentally is indeed 1.4 nm reported in Fig. 3. Other shorter values were acquired in the $k(Z)$ curve showing always the same trend, i.e. a characteristic decay of the $k(Z)$ curve as well as the observed 0.2-0.3 nm periods. Concerning the maximum lifting height, we think that the folded conformation of the ssDNA limits the experiment. In fact, lifting such oligomers with the AFM requires to not only peel off the structure but also unzip it over the surface. This is particularly well-observed in the provided Supp Info movies. Because the tip-molecule bond is rather weak and the adhesion of the folded ssDNA on gold quite strong, only incomplete lifting have been experimentally done. The 1.4 nm value is likely related to the way the structure is folded. However, it is difficult to extract a clear trend since numerous conformation coexists. This would require a profound statistical analysis which is beyond the scope of our work.

To clarify this point, we discussed this point in the main text:

“Experimentally, only partial lifting of single ssDNA oligomers either self-assembled or individually adsorbed (Supp. Info Fig. S7) could be achieved. Thus, it appears that the folded ssDNA conformation and its strong adhesion on gold are limiting factor of the experimental lifting. Indeed, desorbing single-folded oligomer with the AFM tip requires not only peeling off the structure from the gold surface but also unzipping it prior to desorption. Each peculiarity of the folding configuration can cause an abrupt increase of the required force to lift the molecule and the rupture of the tip-molecule bond.”

Concerning future measurements that could allow complete ssDNA lifting, we modified the conclusion as follow:

“ In future experiments, we will also focus on controlling the folding characteristics of the ssDNA using, for example, patterned surfaces to linearly adsorb oligomers. This may enable their complete lifting and the statistical analysis of their detachment. Another alternative is to perform measurements on less adhesive surfaces.”

On page 8, last paragraph, is presented a comparison with other works on molecules desorbed from surfaces with similar experimental procedures. The conclusions are reasonable, but the comparison is addressed in a disorganized and superficial way that should be improved. The same remark applies to the comparison discussed in the Conclusions.

We thank the referee for the comment. We tried to clarify the comparison of the lifting with the previous experiments as well as the conclusion. We modified the section “ssDNA cryo-force spectroscopy” as follow:

“There, the k maxima ($\approx 0.4 \text{ N.m}^{-1}$) were constant during retraction. The process ended at a distance corresponding to the number of monomers initially identified by STM on the surface. In that case (and also for graphene nanoribbons 35), the much stiffer repeat units weakly adhere to the substrate, allowing nearly frictionless sliding and a complete chain detachment. The oligomer backbone is more flexible and most bases are strongly bonded to the gold surface. As shown in our MD simulation (snapshot of Fig. 3e and supplementary movies), the cytosine basis that remain adsorbed do not slide during lifting [34,35]. As a result, the lifted segment between tip and sample becomes essentially straight and inclined as the tip-sample separation increases. This sequential base detachment, similar to peeling off an adhesive tape, also reflects the strong adhesion of adsorbed cytosine bases.”

The Conclusion has been also modified as follow:

“The present work relies on closely matched scanning tunneling and force measurements, and computer simulations. The first part addresses the adsorption and self-assembly of single-strand DNA cytosine oligomers spray-deposited on Au(111) at the sub-nanometer level. Both theory and experiment showed folded ssDNA conformations arising from the first step of the adsorption. Later, the unfolding of the ssDNA on the surface is not possible upon annealing. In the second part, the mechanical response of single ssDNA oligomers lifted from the gold surface is investigated at the same level using cryogenic force spectroscopy. Multi-stage detachment is inferred, similar to peeling off an adhesive tape, and reflects the strong adhesion of adsorbed ssDNA bases on gold. We extracted high values for the initial stiffness measured during lifting ($\sim 15 \text{ N.m}^{-1}$) as well as the comparable pinning stiffness obtained from numerical calculations and intrinsic stiffness per repeat unit of fully stretched ssDNA ($\sim 33 \text{ N.m}^{-1}$). This last value corresponds to the stiffness of the fully stretched 20-cytosine oligomer having a maximum length of 12 nm when subject to an average tension of $\sim 2 \text{ nN}$. This load is one to two orders of magnitude larger than the one applied at room-temperature with single-molecule force spectroscopy on hundreds of nanometer long ssDNA. [12,14–17,41]

A drawback of the present system for experimental lifting is the complex folding of the ssDNA oligomers in comparison to polymeric systems in similar conditions [34,35]. This limits the lifting heights to only a fraction of the extended length. Comparing to $k_{\text{pin}} \approx 0.7 \text{ N.m}^{-1}$ obtained for polyfluorene chains, [34] a complete detachment of ssDNA might be achieved by using end linkers allowing to reinforce the bond between the oligomer and the AFM tip in UHV. Although more difficult than solution chemistry, this task appears feasible in view of the successful detachment of PTDCA from Au(111) following contact to a carboxylic oxygen atom. [33] In future experiments, we will also focus on controlling the folding characteristics of the ssDNA using for example atomically patterned surfaces. That way, linearly adsorbed oligomers may enable complete lifting and possibly a statistical analysis of their detachment. Another alternative is to perform measurements on less adhesive surfaces.”

In “ssDNA cryo-force spectroscopy” section, sentences are often very long without any punctuation and is very difficult to read and understand the text.

We thank the referee for this comment. We have changed the wording of that section following the referee's suggestions. The modifications are highlighted in red in the main manuscript.

In the Methods section is written that CHARMM-METAL FF was used to describe the gold surface. This FF is not able to account for the dynamic polarization of the metal atoms in the Au surface. In the case of DNA, which carries a e charge for each nucleotide, this effect could be significant. Can the author comment about this point?

Using this force field, we observe that both the conformations of the molecule are in agreement with previous works (see ref. 37 of the manuscript) and also that stiffness variations during the chain lifting process is in good agreement with our experimental results. Furthermore, we also observe that the self assembly of the molecules is in agreement with our experimental findings, both from a structural point of view and from the temperature required to achieve these large assemblies. Note that this former property is highly sensitive to the interaction between the surface and the molecule. Therefore, CHARMM-METAL FF seems to be correctly describing the interaction between gold and ssDNA, in agreement with previous observation for other charged peptide chains (see ref. 14 of the methods section). Thus our results suggest that dynamic polarization effects, although not explicitly present in the used FF, seem to be either accounted for in an effective manner through the LJ potential or it has a weak contribution.

We may rationalize the findings above considering that the CHARMM-METAL FF has been parametrized against DFT calculations to reproduce not only correct adsorption configurations but

also adsorption energies and dynamical/mechanical properties of the gold surface. The comparison of the computed adsorption energies has been previously validated through different experiments of peptides adsorption showing a qualitative accuracy in the 10-20% range (a comprehensive list can be found in the review article of ref. 14 of the methods section). Interestingly, our simulations also show that the stiffness of the gold-molecule interaction is in good agreement with the experimental values. As for the polarizability effects, this force field incorporates them through the Lenard-Jones parameters – in a static manner. Still, it is worthwhile mentioning that the image charges occurring in the metal atoms as a result of an adsorption of charged bio-polymers, have been shown to play only a marginal contribution to the total adsorption energy (some examples are provided in the aforementioned review of ref.14).

Alternatively, one could also consider other force fields such as GOLP or GOLP-CHARMM. In these models, the gold atoms are frozen and they incorporate a dynamic polarizability through freely rotating rod-like dipoles. When sliding a molecule over the surface, this might become an issue. The sliding of the molecule occurring during the lift process occurs in a stick-slip fashion, i.e. a stick phase where the atoms of the molecule are initially pinned to the atoms of the surface; and a slip phase where by accumulating enough tension through a pulling force one may slide. This means that our model must consider the transfer of momentum from the molecule atoms to the surface and also the vibrations of both the surface and molecule atoms which will facilitate the sliding. In the aforementioned dynamic polarizable force field, since the atoms of the surface are frozen, there is neither conservation of momentum nor thermal vibrational energy. Therefore, this would lead to an incorrect description of the physical processes of slip/stick at this interface. If we would unfreeze the gold surface, the gold atoms would all collapse due to the lack of a proper description of their internal degrees of freedom. Taking this into account, it is not clear that the gain introduced using these complex force fields (e.g. better adsorption configurations or a slight gain on the accuracy of adsorption energies) would compensate the pitfall of using a frozen atom surface in a dissipative process such as the sliding of molecules over the surface. This is what ultimately dictated our choice on using CHARMM-METAL FF.

In Section S4 in the Supporting Information it is very difficult to understand which calculations are performed and many parameters are missing.

We fully agree with the referee and thank him/her for the comments. We have extended the methods section so that all the necessary parameters and protocols are completely detailed to the level necessary to reproduce all of our simulations. Additionally, we have also rewritten the section S4, so that also the details of these additional tests and their purposes become more clear. We are very grateful that the referee brought these aspects that helped us to improve our manuscript.

Figures S3 and S4 show the Na⁺ ions solvated in the large cell of water, but when water molecules are removed from the cell apart within 1nm from DNA, are Na⁺ ions all packed at this short distance from the molecule? Does this choice affects the results of the simulation?

In all our simulations, all 19 Na⁺ ions are present. This has now been made clear thanks to the rewriting of the methods section motivated by a previous comment of the referee.

In all our simulation, we are using Particle-Mesh-Ewald summation to compute the non bonded interactions. With this set up, implemented in AMBER and other codes, it is not possible to perform an MD simulation for a charged system, because the spurious interaction between periodic images cannot be removed. Therefore, we cannot determine what would be the influence of not including all the Na⁺ counter-ions. Nevertheless, it is reasonable to assume that a charged molecule such as the ssDNA remains charge compensated in the conditions of these experiments (the dissolution of the spray does contain neutralizing ions which are also pumped into the chamber during the spray

deposition). Also, even after annealing up to 500K we observed in our MD simulations the mobility of the Na⁺ ions with respect to the ssDNA molecule was negligible, as they preserved their position within the chain and displaced solidly over the surface with the ssDNA strands. At variance with this, in the fully water embedded simulation we observed that Na⁺ ions often went away from the strand, as shown in Fig. S3. This higher ionic mobility may be understood in terms of a better screening of the electrostatic interactions provided by the solvent medium. As shown in Fig. S4 (t=0ns), the configuration chosen to start the ssDNA droplet in vacuum MD simulation contained within 1~nm all the neutralizing counter-ions. Therefore, in the droplet simulation in vacuum all the Na⁺ counter-ions are included. Furthermore, in this case, we observed that after an initial 10~ns transient all ions remained inside the droplet in a stable equilibrium position as shown in Fig. S4.

We thank that the referee brought this point to our attention. In order to clarify it we have made explicit the selection criteria in the methods section.

In paragraph “Simulated ssDNA diffusion and self-assembly on Au(111)” it is written “Note that a much longer simulation time was required”. To do what?

We have just removed that sentence. We appreciate that the referee brought this into our attention.

Minor issues:

Figures of computational snapshot are small and printed with low resolution (e.g. Figure S3)

We now included images with higher resolutions.

In the first panel of Figure S3 and S4 it is not clear what is the empty cube surrounding the DNA

The empty cube surrounding the ssDNA simply represents the dimensions of the simulation box. We thank that the referee brought this to our attention and we have corrected the captions of Fig. S3 and S4 so that this is clear.

We acknowledged the referee for the precious comments helping us to improve your manuscript. We substantially revised the manuscript accordingly.

Reviewers' Comments:

Reviewer #2:

Remarks to the Author:

I sincerely thank the authors for the efforts of revision. Now I understand that "the determination of the overall ssDNA conformation is rather difficult".

I think additional data and discussion greatly strengthen the paper.

I therefore recommend acceptance.

Reviewer #3:

Remarks to the Author:

In the new version of the paper of Pawlak et al., the authors greatly improved the clarity of the paper and in particular of the calculations performed. The comments in the remarks are detailed and convincing and the authors obtained nice results both experimentally and computationally. Nevertheless I still think that concerning the comparison between experiments and simulations, some improvements could be obtained.

Concerning the answer about the presence of the counterions in PBC simulations where PME is used, I would like to underline that counterions are not mandatory to neutralize the charge of the system. Charged systems can be simulated with a neutralizing background, a rather standard procedure, that would allow to study the effect of Na⁺ ions in determining the geometry of ssDNA. In the revised version of the paper the described simulation procedure is clear and I think that including the ions in the "drop" of water is a reasonable choice. Nevertheless the authors should be aware of this possibility, particularly when discussing the comparison between simulations and experiments. I think that the comparison between simulations and experiment is overall reasonable and that simulations support experimental results. Nevertheless there are some inaccuracies in the comparison that could be improved, considering also the fact that the authors already performed the majority of the simulations needed.

The authors say that they perform an annealing procedure from 300 K to > 500. The comparison between AFM images and classical MD calculations is found approx at 440 K, when the water molecules are already desorbed from the gold surface but there is no aggregation. Nevertheless the comparisons in Figure 1 c) and d) are with a geometry obtained with a simulation at 300K with water and counterions. I think that the comparison should be done with the molecule adsorbed on the surface without water at 440 K (and maybe without Na⁺ ions, depending on what they think about the experimental system). From the supporting informations it seems likely that the comparison should be not negatively affected by this difference in the simulation parameters. There is a specific reason for which the authors chose differently?

On the same topic, concerning the answer to Reviewer 2 where they discuss the presence of water molecules at 300 K to explain the shape and dimension of clusters in AFM images, at point 2 the authors state that "According to our numerical simulations (Fig. 2a), the ssDNA oligomers adsorb on Au(111) with their hydration layer". They claim that this result supports their statement that water molecules are present on the surface at 300 K after the deposition. The statement seems reasonable to me and in agreement with the explanation of the experimental procedure and of the results of the annealing process. Nevertheless I would like to point out that I don't think that the water droplet simulation as it is described can be used to strengthen this assumption. I feel that the presence of

water on the surface is a consequence of the experimental procedure and that the starting configuration of the simulation is properly chosen to reproduce the experimental configuration: with the water already interacting with DNA and starting already close to the surface, the most probable outcome was to have it adsorbed on gold. Moreover, in the same answer, the authors say: "All crucial steps, i.e. nano-droplet deposition, water desorption, ssDNA diffusion and self-assembly were also confirmed by numerical simulation". Is really water desorption reproduced with a simulation at 440 K? Water simulation is known to be tricky and such a result would be worth to mention, particularly in view of the comparison with experiments.

In the method part the description of the steered MD simulation is missing. Is it performed in vacuo? With or without ions? Same question arises for the simulations describing the aggregation of the two ssDNA molecules: at 440K and 500 K the simulation should probably be done without water or, referring to previous paragraphs, featuring the desorption of water molecules. These simulations should try to match the experimental conditions or discuss why the differences don't influence the result.

This is particularly true for the simulation studying the aggregation at 300K. The experimental condition from the AFM image is of different ssDNA clustered together with water molecules. It seems to me that in the simulation at 300 K this is not the case, as the two molecules are 3 nm distant, which means that there is vacuo in between them. To reproduce the experimental condition they should be both inside a larger drop of water. Maybe this difference can affect the result on the diffusion and aggregation? In Figure 1 a) and b) the ssDNA molecules seem grouped in clusters together with water molecules, while in figure 1 c) at 440 K they seem more evenly distributed. The authors are sure that no diffusion of ssDNA happened during water desorption?

Concerning the answer and discussion about the choice of the Force Field and the agreement with previous computational work on cytosine adsorbed on gold in vacuo, the authors may want to cite [Rosa et al. - Langmuir, 2018 DOI:10.1021/acs.langmuir.8b00065] where adsorption on single nucleobases on gold is discussed, with adsorption configurations of nucleobases in solution in agreement with the configuration adopted by ssDNA in their work.

Reviewers' comments:

Reviewer #2 (Remarks to the Author):

*I sincerely thank the authors for the efforts of revision. Now I understand that "the determination of the overall ssDNA conformation is rather difficult".
I think additional data and discussion greatly strengthen the paper.
I therefore recommend acceptance.*

We thank referee#2 for the carefully reading our manuscript and acknowledge her/his very positive feedback.

Reviewer #3 (Remarks to the Author):

*In the new version of the paper of Pawlak et al., the authors greatly improved the clarity of the paper and in particular of the calculations performed. The comments in the remarks are detailed and convincing and the authors obtained nice results both experimentally and computationally.
Nevertheless I still think that concerning the comparison between experiments and simulations, some improvements could be obtained.*

We are very grateful for the large effort put in the thorough constructive criticism provided by the referee. Furthermore, we appreciate that the referee recognized the interest of the current work. In what follows we carefully address all the issues raised by the referee.

Q1) Concerning the answer about the presence of the counterions in PBC simulations where PME is used, I would like to underline that counterions are not mandatory to neutralize the charge of the system. Charged systems can be simulated with a neutralizing background, a rather standard procedure, that would allow to study the effect of Na⁺ ions in determining the geometry of ssDNA. In the revised version of the paper the described simulation procedure is clear and I think that including the ions in the “drop” of water is a reasonable choice. Nevertheless the authors should be aware of this possibility, particularly when discussing the comparison between simulations and experiments.

We appreciate that the referee raised this question. In principle charged systems can be treated using other MD packages, such as GROMACS but in our current simulation setup (AMBER14) that would require changing the code. However, for experimental reasons provided in the revised version (in Supplementary Section 1 and the Methods section), the ssDNA oligomers were charge-compensated both before spraying and after deposition on the metallic gold substrate. Furthermore, the good agreement obtained for the final adsorption configurations indicates that our simulations suitably describe the experimental observations. Indeed, a deficit or excess of more than one counter-ion would likely result in more extended adsorption configurations due to the strong electrostatic repulsion between uncompensated phosphate groups of the ssDNA backbone or attached Na⁺ ions, respectively.

Modifications motivated by Question 1

> **Modification #1**, page 19 of Manuscript (methods section)

“... The single-strand 20-mer DNA molecules were purchased from Microsynth (Switzerland). The solution provided was subjected to High Performance Liquid Chromatography (HPLC) thus ensuring that only organic molecules present were 20--cytosine ssDNA oligomers. Subsequently, dialysis was performed in the presence of an excess of NaCl which guaranteed that any other salts generated during synthesis are exchanged by NaCl. The delivered solution contained 14.7 nM of ssDNA and a NaCl concentration sufficient to provide more than twice the required amount of Na⁺ neutralizing counter-ions. This solution was further diluted to a concentration of around 1 nmol.ml⁻¹ in water and then sprayed in UHV ...”

> **Modification #2**, page 2 of Supplementary Information

“... The spray is formed by applying a high voltage between the syringe and the grounded inlet capillary (inner diameter 750µm). This leads to the formation of a jet of multiply charged droplets that are accelerated then pass through the differential pumping system. Solvent evaporation causes coulomb fission which results in smaller droplets with increasing concentration of ssDNA and salt, down to nano-droplets containing single molecules and eventually gas-phase ionized complexes^[1] ^[2]. During ESD, the base pressure of the preparation chamber reaches $p = 10^{-7}$ mbar. Typical high voltage values ~ 1.5 kV are adjusted to control the spray quality by visually observing the overall spray shape during deposition. In solution solvated DNA and ions interact via screened electrostatic interactions, and the ions form a neutralizing space-charge sheath around each DNA molecule. This also happens in the interior of large droplets but, in the final nanodroplets, solvation shells may overlap, so that Na⁺ counter-ions become strongly bound to nearby PO₄⁻ ions of the ssDNA backbone. Before deposition these nanodroplets carry surface charges, most likely excess Na⁺ ions, as indicated by mass spectra of proteins electro-sprayed from similar solutions by other investigators^[2]. Upon impinging on the metallic surface, however, those excess ions are very likely neutralized since the gold substrate is grounded.”

> **Modification #3**, page 3 of Supplementary Information

“... Another important property of ssDNA is that each phosphate carries one negative charge which is, however, compensated by ions from the environment. In the case of dsDNA, on average 90% of the solute cations (Na⁺) are accumulated, whereas only 10% of anions (Cl⁻) are depleted within a few nanometers[6]. The combined effects ...”

> **Modification #4**, page 12 and Fig.S8 of Supplementary Information

“... Finally, in all MD simulations we found that after adsorption Na⁺ counter-ions preserved their relative positions with respect to the ssDNA backbone. Interestingly, this behavior persisted even at 500K (see Fig.S8d) when both molecules slightly change their conformations while diffusing. Concurrently, in the experiments we observe that after 440 K annealing the prevalent features display a similar length of about 4nm (see Fig.1a and Fig.S1). This observation supports our assumption that they correspond to dehydrated ssDNA molecules with the same number of neutralizing ions. Indeed, a charge undercompensated ssDNA would adopt a more extended conformation to minimize the Coulomb repulsion between uncompensated negative phosphate groups.”

Q2) I think that the comparison between simulations and experiment is overall reasonable and that simulations support experimental results. Nevertheless there are some inaccuracies in the comparison that could be improved, considering also the fact that the authors already performed the majority of the simulations needed.

We thank the referee for his/her positive evaluation on the agreement between the experiments and simulations. Below we provide further clarifications (and corresponding modifications) which resolves the mentioned issues.

Q3) The authors say that they perform an annealing procedure from 300 K to > 500. The comparison between AFM images and classical MD calculations is found approx at 440 K, when the water molecules are already desorbed from the gold surface but there is no aggregation.

Nevertheless the comparisons in Figure 1 c) and d) are with a geometry obtained with a simulation at 300K with water and counterions. I think that the comparison should be done with the molecule adsorbed on the surface without water at 440 K (and maybe without Na⁺ ions, depending on what they think about the experimental system). From the supporting informations it seems likely that the comparison should be not negatively affected by this difference in the simulation parameters. There is a specific reason for which the authors chose differently?

The referee made us realize that we did not clearly specified which atomic configuration is used to compare simulations results with experimental ones. The comparison in Fig.1 is in fact done using a structure obtained at 400K in vacuum conditions, thus in conditions similar to experimental ones as requested by the referee. From our MD simulations we may conclude that the slight temperature difference between experiments and simulations, i.e. 440K vs. 400K, has no significant implications on the final adsorption configuration. This is shown in the new Fig.S6 and Fig.S7 where we observe that oligomer length is independent of the hydration, starting configuration and temperatures up to 450K. In addition, all simulations reported in this study include neutralizing counter-ions (see modifications in Question1). The present question motivated us to summarize all the simulations performed and their relationships in a new Supplementary Section 3 and to renumber the following ones.

Modifications motivated by Questions 2/3

> **Modification #1**, page 4 of Manuscript (Fig.1 Caption)

“Figure 1: Evolution of ssDNA morphologies as a function of Au(111) annealing temperature. (...) c, Top view of a representative ssDNA structure on Au(111) obtained from MD simulations performed in vacuum at 400K (stage 4 of simulation protocol – see Methods). d, STM image ... ”

> **Modification #2**, page 5 of Manuscript

“(...) Their overall size also corresponds to the structure of a folded oligomer adsorbed on Au(111) as systematically predicted by our simulations performed under different adsorption conditions (see Supplementary Section 5) and superimposed as top views in Fig1b, d and e. (...)”

> **Modification #3**, page 6 of Manuscript (Fig.2 Caption)

“Figure 2: Molecular dynamics simulations of adsorption and diffusion of ssDNA over Au(111). a, Side- and top views of a small water droplet containing one ssDNA oligomer getting adsorbed on the gold surface (stage 3 of simulation protocol). Water is represented using a transparent surface; scale bars = 1 nm. b, 500 ns--MD simulation of two oligomers assembled by diffusion at 500 K (stage 5 of simulation protocol – see Supplementary Sections 3 and 6). At 500~K, both oligomers start diffusing, thus favoring self-assembly assisted by intermolecular interactions ; scale bars =1 nm.”

> **Modifications #4**, page 7 of Manuscript

“... allowed to adsorb onto an unreconstructed Au(111) surface at room temperature (see Supplementary Section 3). We first used the ssDNA conformation... ”

> **Modification #5**, page 22 of Manuscript (Methods)

*“**Simulations protocols.** In total we have performed 8 different MD simulations each labeled as a stage, all summarized in Supplementary Section 3. Here we briefly outline each stage. Stage 1: ssDNA oligomer fully embedded in water at 300K. Stage 2: ssDNA inside a water droplet in vacuum at 300K. Stage 3: Adsorption of a water droplet containing one ssDNA onto a Au(111) surface at 300K. Stage 3a: Water evaporation at 450K from the same adsorbed droplet. Stage 3b: Evolution of a fully dehydrated ssDNA adsorbed on Au(111) in vacuum at 400K. Stage 4: Adsorption of a single DNA strand in vacuum onto Au(111) at 400K. Stage 5: Diffusion and self-assembly of two ssDNA molecules adsorbed on Au(111) in vacuum at 400K and 500K. Stage 6: Lifting a ssDNA molecule adsorbed on Au(111) in vacuum at 5K. In all simulations 19 Na⁺ counterions per ssDNA molecule were included to ensure overall charge neutrality.”*

> **Modification #6**, pages 4-7 of Supplementary Information (a whole new section)

“3. Simulation Protocols

All simulations used in this work are schematically summarized in Fig.S3. They were performed in constant N and T ensembles where a Langevin thermostat maintained the average temperatures specified in Fig.S3. Stage 3b, stage 4 stage 5 and stage 6 simulations were performed in vacuum. In all the others water molecules were considered as specified below. In all simulations we consider one 20-cytosine ssDNA oligomer and include 19 Na^+ counter-ions to ensure overall charge neutrality.

Figure S3 | Chart describing the different MD simulations performed and their relationship. The arrows indicate that the final configuration of a given stage is used as a starting configuration of the connected stage, e.g. the final configuration of stage 3 is used as a starting configuration of stages 3a and 3b. Gold atoms are represented as yellow Van-der-Waals (VdW) spheres; ssDNA as liquorice; transparent cyan Connolly surfaces enclose the water molecules present in the system; and purple VdW spheres represent sodium counter-ions.

Stage 1: 20-cytosine ssDNA equilibrated in aqueous solution. Here we start from a 20-cytosine DNA single strand in the canonical B-form generated using the NAB software available in AMBERTOOLS put it in the center of a $11.6 \times 11.6 \times 11.6 \text{ nm}^3$ simulation box filled with 41190 water molecules and 19 sodium counter-ions. After an energy minimization to avoid steric clashes in the MD simulation we performed a 100ns long NPT run (...)

To avoid an unnecessary lengthy reply, please refer to the new Supplementary Section 3 for a description of all simulation stages. Above you find a new figure included there that summarizes all simulations performed and their relationships.

Q4) On the same topic, concerning the answer to Reviewer 2 where they discuss the presence of water molecules at 300 K to explain the shape and dimension of clusters in AFM images, at point 2 the authors state that “According to our numerical simulations (Fig. 2a), the ssDNA oligomers adsorb on Au(111) with their hydration layer“. They claim that this result supports their statement that water molecules are present on the surface at 300 K after the deposition. The statement seems reasonable to me and in agreement with the explanation of the experimental procedure and of the results of the annealing process. Nevertheless I would like to point out that I don't think that the water droplet simulation as it is described can be used to strengthen this assumption. I feel that the presence of water on the surface is a consequence of the experimental procedure and that the starting configuration of the simulation is properly chosen to reproduce the experimental configuration: with the water already interacting with DNA and starting already close to the surface, the most probable outcome was to have it adsorbed on gold. Moreover, in the same answer, the authors say: “All crucial steps, i.e. nano-droplet deposition, water desorption, ssDNA diffusion and self-assembly were also confirmed by numerical simulation”. Is really water desorption reproduced with a simulation at 440 K? Water simulation is known to be tricky and such a result would be worth to mention, particularly in view of the comparison with experiments.

We agree that the presence of water molecules is a consequence of the experimental procedure. In accordance, we have reformulated the description our droplet adsorption simulation (stage 3). See Mod#1 below.

Concerning water desorption, our previously not mentioned simulation at 450K (see Mod#2 below) provide evidence for the evaporation of water-molecules above and aside the adsorbed ssDNA oligomer. Remarkably, this happens without altering its configuration. Note that since the 450K simulation is performed in the NVT ensemble, the desorbed water molecules float in vacuum or eventually adsorb elsewhere on the gold slab. This can also be understood in light of the very recent work mentioned by the referee which is now properly cited (see Q5 below). According to this publication, adsorption of single nucleotide bases on Au(111) in water at 300K ultimately leads to direct contact between cytosine and the gold surface. Careful examination of Fig.2a shows that, following droplet adsorption in our stage 3 simulation at the same temperature, no water appears between the whole ssDNA oligomer and the gold surface. Thus our stages 3, 3a and 3b simulations not only strongly support our ‘direct contact claim’ but they are also in agreement with experimental observations after 440K annealing where ssDNA is no longer surrounded by water molecules as seen in Fig.1b at 300K. Achieving full dehydration in a prolonged simulation at 450K seemed an unnecessary and complex task which was left unmentioned in our originally submitted manuscript.

Q5) This was originally question#6. We placed it here due to its relation to Q4. Concerning the answer and discussion about the choice of the Force Field and the agreement with previous computational work on cytosine adsorbed on gold in vacuo, the authors may want to cite [Rosa et al. - Langmuir; 2018] where adsorption on single nucleobases on gold is discussed, with adsorption configurations of nucleobases in solution in agreement with the configuration adopted by ssDNA in their work.*

We sincerely appreciate that the referee brought this work to our attention. This reference not only strongly supports one of our important findings but also provides additional information concerning the adsorption mechanism and lowest free energy configurations of nucleotide bases on Au(111) in vacuum and in water at 300K that complement our results obtained under the same conditions. We have properly cited in the modifications detailed below.

Modifications motivated by Questions 4/5*

> **Modification #1**, page 5/6 of Manuscript

“MD simulations of ssDNA adsorption, dehydration, diffusion and self assembly on gold

Instead of simulating the complex electro-spray deposition processes which start and end up in charge-neutral species (Supplementary Section 1), in all our simulations we considered a single ssDNA oligomer together with charge-compensating counter-ions. For a comparison with Fig.1b, a 20-cytosine ssDNA oligomer generated from the canonical B-form¹ (Supplementary Fig. S2) together with 19 Na⁺ ions was inserted into a water droplet, equilibrated, then allowed to adsorb onto an unreconstructed Au(111) surface at room temperature. ...”

> **Modification #2**, page 9/10 of Supplementary Information & Fig.S6

“In order to study the influence of water removal on the final adsorption configuration of the oligomer, we performed two additional simulations. In stage3a, we anneal up to 450K the adsorbed droplet obtained at the end of stage3. The snapshots shown below in Fig.S6 reveal that during the 100ns of annealing water starts to evaporate above the adsorbed ssDNA and from the gold surface in its vicinity, whilst keeping the oligomer conformation unaltered. This is compatible with a strong interaction between ssDNA and gold, in particular because most nucleotide bases are in direct contact with the gold surface already prior to annealing, as described in the main text. Since we are performing our MD simulations in the NVT ensemble, the total number of molecules and ions in the system remains constant. The water molecules near the ssDNA are either floating in the simulation box or condensed on the constrained bottom layer of gold in the adjacent box owing to periodic boundary conditions (see Methods).

In stage3b we considered same starting adsorbed ssDNA configuration with neutralizing counter-ions but removed all water molecules. Letting the system evolve at 400K during 100ns we obtain the final results shown in Fig.S6 (right) and in greater detail in Fig.S7b. They clearly show that removing all water molecules has little effect on the final adsorption configuration of the ssDNA oligomer. We thus conclude that dehydration occurs with little influence on the compact folded conformation of the adsorbed ssDNA oligomer, in particular its width and overall length.”

Figure S6 | Effect of water removal on the final conformation of ssDNA adsorbed on Au(111). From left to right: Initial and final atomic configurations of stage 3a (450K annealing of the pre-adsorbed droplet – see Sec.3 of Sup.) compared to the final configuration obtained using the initially dehydrated adsorbed oligomer (stage 3b). The atomic representation used is the same as in Fig.S3. Scale bar = 1nm.

Modification #3, page 10 of Supplementary Information

“(…)Note that only a few bases protrude while the rest lie essentially parallel to the surface at an average distance of 0.27 nm from the Au(111) surface layer. Similar adsorption geometries were predicted for an isolated cytosine base in vacuum and in water at 300K by using a different force field [Marta Rosa 2018] fitted to Van-der-Waals density functional calculations by the same group. (…)”

> **Modification #4**, page 7 of Manuscript

“…To test the effects of dehydration and thermal annealing on the final adsorption configuration, starting from the adsorbed hydrated ssDNA shown in Fig.2a we simulated: (i) partial water evaporation by annealing at 450K (stage 3a, see Supplementary Fig.S6), (ii) the effect of removing all water molecules (stage 3b, see Supplementary Fig.S7) and (iii) by simulating the adsorption of a single DNA strand at 400K without any water molecules (stage 4, see Fig1c and Supplementary Fig.S7). All adsorption simulations, including stage 3 (hydrated ssDNA at 300K) led to very similar folded adsorbed conformations ~4 nm in size, in good agreement with the prevalent experimentally observed structures after annealing at 440K, as shown in Fig.1.

The systematic nature of the findings can be traced back to the direct ssDNA/Au(111) contact already established at 300K before dehydration. Concurrently with this direct contact ssDNA/Au(111) contact, the interaction of cytosine bases with gold also induces a systematic flattening of the ssDNA structure. Indeed, most bases lie nearly parallel to the surface, similar to optimum adsorption structures for single nucleotide bases on Au(111) computed in vacuum using a nonlocal VdW density functional [Rosa 2013] and very recently by MD simulations in vacuum and in water at 300~K [Rosa 2018].”

Q6) + In the method part the description of the steered MD simulation is missing. Is it performed in vacuo? With or without ions?*

+ Same question arises for the simulations describing the aggregation of the two ssDNA molecules: at 440K and 500 K the simulation should probably be done without water or, referring to previous paragraphs, featuring the desorption of water molecules. These simulations should try to match the experimental conditions or discuss why the differences don't influence the result.

In the modifications motivated by Q3 one may find a new section 3 in the Supplementary material and changes in the methods section. These clarify that the steered MD simulations were performed with neutralizing ions, and in vacuo at 5K. Additional details about the determination of the pulling force, extraction of the force gradient and its relation to the measured frequency shift are provided in the modifications detailed below. The self-assembly simulations at 400 and 500K were also performed in vacuum in agreement with referee's expectations.

This is particularly true for the simulation studying the aggregation at 300K. The experimental condition from the AFM image is of different ssDNA clustered together with water molecules. It seems to me that in the simulation at 300 K this is not the case, as the two molecules are 3 nm distant, which means that there is vacuo in between them. To reproduce the experimental condition they should be both inside a larger drop of water. Maybe this difference can affect the result on the diffusion and aggregation? In Figure 1 a) and b) the ssDNA molecules seem grouped in clusters together with water molecules, while in figure 1 c) at 440 K they seem more evenly distributed. The authors are sure that no diffusion of ssDNA happened during water desorption?

We agree that the simulations of diffusion performed in vacuum at 300K (now omitted to avoid confusion) did not reproduce the exact experiment conditions. The lack of diffusion at 300K is indeed expected from the lack of diffusion at 400K and from the direct contact between ssDNA and Au, which is already achieved at 300K in our simulations of water nano-droplet adsorption (see Fig.2a).

At 300K it is difficult to tell if some droplets seen in Fig.1b contain more than one ssDNA molecules. Nevertheless, after annealing to 440K, in the experiments we observe prevalent features reasonably assigned to single molecules (see Fig.1). We therefore focused our attention on the comparison between images of samples annealed at 440K or 500K and simulations of dehydrated, charge-compensated ssDNA oligomers at 400 and 500K.

Modifications motivated by Question 6*

> **Modification #1**, page 9 of Manuscript

“... From the recorded variations of the noise-averaged normal force $\langle F(Z) \rangle$ (Fig.3c), the effective stiffness was extracted as $k = d\langle F(Z) \rangle / dZ$ (Fig.3d), Z being the distance between the tip atom and pulled P atom at $t = 0$. (...)”

> **Modification #2**, page 20 of Manuscript (**STM/AFM microscopy in Methods**)

“... The constant-height AFM images were acquired with CO-terminated tips using the non-contact mode by driving the free prong on resonance while maintaining a constant tip oscillation amplitude $A = 50 \text{ pm}$. For such small A the frequency shift induced by a smoothly varying force acting on the tip is to a good approximation proportional to the gradient k (effective stiffness) of the conservative force along the oscillation direction (perpendicular to the sample surface). The signal/noise ratio is then nearly optimal [Giessibl 2013].”

> **Modification #3**, p. 22 of Manuscript (**Molecular Dynamics simulation details in Methods**)

“In the steered MD simulations results shown in Fig. 3 and Supplementary Fig. S10, the conservative force $F(Z)$ was computed as $k_{\text{tip}}(Z - Z_P)$ and its thermal average $\langle F(Z) \rangle$ approximated by a running average over (100ps) an interval adjusted to obtain a smooth dependence without distortions except close to force drops. The resulting $d\langle F(Z) \rangle / dZ$ can be safely²⁰⁻²² compared to the measured effective stiffness k between negative slips in the intervals where k is positive.”

> **Modification #4**, Supplementary Fig. S8

We removed the results of the simulation of 2 ssDNA molecules in vacuum at 300K.

> **Modification #5**, page 8 of Manuscript

“The effect of thermally assisted diffusion of ssDNA on Au(111) has been investigated with 500ns long MD simulations of two adsorbed ssDNA at temperatures of 400 K and 500 K (see stage 5 of simulation protocol in Methods). Diffusion and coalescence of the ssDNA oligomers is observed only at $T = 500 \text{ K}$ (see Fig.2b and Supplementary section 6) which is in agreement with the experimental data. (...)”

> **Modification #6**, page 12 of the Supplementary Information (section 6)

“In our experiments (see Fig.1), we observed molecular assemblies with lengths over 40nm only after annealing at 500K. Therefore, and in agreement with our simulations (Fig.S8), there is no long-range diffusion of dehydrated ssDNA molecules below 500K. As discussed in the main text, thermal annealing has two direct consequences: water evaporation and ssDNA diffusion. As shown in Fig.1, annealing to 440K results in desorption of water molecules but no long-range diffusion of single ssDNA oligomers identified as the prevalent structures with an overall length $\sim 4 \text{ nm}$ matching the simulated ones. Nevertheless, when comparing the STM images after annealing at 340K and 440K shown in Fig.1a, we observe a higher ssDNA density at 440K. This difference most likely results from the uneven surface coverage on larger scales resulting from electro-spray deposition. Indeed, before each annealing step, the tip must be retracted thus making it impossible to image the same region of the sample afterwards, owing to thermal drift during annealing.”

> **Modification \$7**, page 12 of the Supplementary Information (section 6)

“Although at 300 and 340K we do not observe long-range diffusion of ssDNA molecules it is instructive to dwell on the possible role of water molecules. Our simulations of hydrated ssDNA (see Fig.2 and Fig.S6) show that the oligomer is in direct contact with the gold surface, i.e. water molecules are only present around or on top of the ssDNA. Based on this direct ssDNA/Au contact, one could argue that water molecules would have a marginal role, if at all, in the diffusion of hydrated ssDNA over that gold surface. In our simulations, we observed that hydrated ssDNA did neither diffuse at 300K (stage 3) nor even during water evaporation (stage 3a).”

Reviewers' Comments:

Reviewer #3:

Remarks to the Author:

In the last version of the paper the authors included many information and details which improve the clarity of the paper, especially concerning the computational part. The results are interesting and worth to be published.

I therefore recommend acceptance.